# Visualizing structural transitions of ligand-dependent gating of the TRPM2 channel

Ying Yin[1], Mengyu Wu[2], Allen L. Hsu [3], William F. Borschel[1], Mario J. Borgnia[1,3], Gabriel C. Lander [2] & Seok-Yong Lee [1]

The transient receptor potential melastatin 2 (TRPM2) channel plays a key role in redox sensation in many cell types. Channel activation requires binding of both ADP-ribose (ADPR) and $Ca^{2+}$. The recently published TRPM2 structures from *Danio rerio* in the ligand-free and the ADPR/$Ca^{2+}$-bound conditions represent the channel in closed and open states, which uncovered substantial tertiary and quaternary conformational rearrangements. However, it is unclear how these rearrangements are achieved within the tetrameric channel during channel gating. Here we report the cryo-electron microscopy structures of *Danio rerio* TRPM2 in the absence of ligands, in complex with $Ca^{2+}$ alone, and with both ADPR and $Ca^{2+}$, resolved to ~4.3 Å, ~3.8 Å, and ~4.2 Å, respectively. In contrast to the published results, our studies capture ligand-bound TRPM2 structures in two-fold symmetric intermediate states, offering a glimpse of the structural transitions that bridge the closed and open conformations.

[1] Department of Biochemistry, Duke University School of Medicine, Durham, NC 27710, USA. [2] Department of Integrative Structural and Computational Biology, The Scripps Research Institute, La Jolla, CA 92037, USA. [3] Genome Integrity and Structural Biology Laboratory, National Institute of Environmental Health Sciences, National Institutes of Health, Department of Health and Human Services, Research Triangle Park, NC 27709, USA. Correspondence and requests for materials should be addressed to S.-Y.L. (email: seok-yong.lee@duke.edu)

The transient receptor potential melastatin (TRPM) ion channel family is a part of the TRP channel superfamily and comprises eight members (TRPM1 to TRPM8) that carry out diverse functions in a variety of physiological pathways[1]. TRPM2 is a calcium-permeable nonselective cation channel that is widely expressed in the nervous, immune, and endocrine systems, and plays a crucial role in warmth and redox-dependent signaling[2–4]. Studies of TRPM2-deficient mice have shown that TRPM2 channels expressed in sensory and central neurons are responsible for sensation of warm temperatures and that they contribute to body temperature regulation[5,6]. TRPM2 has also been found to be activated by reactive oxygen species (ROS), which indicates its key role in $Ca^{2+}$ signaling that leads to chemokine production, insulin secretion, and cell death under oxidative stress[7–11].

Extensive electrophysiological studies have shown that activation of TRPM2 by ROS is caused by an increase of intracellular ADP-ribose (ADPR), a TRPM2 agonist, and that both ADPR and $Ca^{2+}$ are required for channel activation[3,12–16]. Notably, TRPM2 contains a putative enzyme domain, called the Nudix Hydrolase 9 Homology (NUDT9H) domain located at the C-terminus, which exhibits a high degree of homology to the mitochondrial ADPR pyrophosphatase NUDT9[17,18]. As such, TRPM2 was initially classified as a channel-enzyme, in which enzymatic activity was believed to be coupled to channel gating[3,17]. However, subsequent studies have since repudiated this mechanism, as the NUDT9H domain lacks enzymatic activity and has been suggested that this domain only serves as a binding site for ADPR[19].

Recently, structures of the zebrafish *Danio rerio* TRPM2 were reported in the ligand-free closed state and in the ADPR/$Ca^{2+}$-bound open state (hereafter referred to as TRPM2$_{closed}$ and TRPM2$_{open}$, respectively)[20]. This study not only revealed the location of the NUDT9H domain but also showed that the $Ca^{2+}$ ion binds in the cavity formed by the voltage-sensor-like domain (VSLD) comprising transmembrane segments 1–4 (S1–S4). Unexpectedly, ADPR was found to bind in the cleft of the melastatin homology domain 1/2 (MHR1/2) and not, as had long been predicted, in the NUDT9H domain. Structural analyses of the closed and open conformations have provided a mechanism for ligand-dependent activation of the channel, wherein binding of ADPR in the MHR1/2 domain triggers substantial conformational rearrangements in the cytoplasmic domain (CD), which are sequentially transduced and propagated to the distal transmembrane channel domain (TMD) to induce pore opening.

In spite of this progress toward understanding ligand-dependent activation of TRPM2, it remains unclear whether these drastic quaternary structural rearrangements occur in a concerted, four-fold symmetric manner. We recently reported that the transient receptor potential vanilloid 2 (TRPV2) channel adopts two-fold symmetric conformations upon resiniferatoxin (RTx)-mediated activation[21,22], and that two-fold symmetric states appear to be associated with ligand-dependent gating of the TRPV3 channel[23]. It is presently unclear if reduced symmetry is associated with gating in other TRP channel families. To address these questions, we determined the cryo-electron microscopy (cryo-EM) structures of the full-length TRPM2 from *Danio rerio* (TRPM2$_{DR}$) in the ligand-free condition (TRPM2$_{DR\_Apo}$), in the presence of $Ca^{2+}$ (referred to as TRPM2$_{DR\_Ca2+}$), and in the presence of both ADPR and $Ca^{2+}$ (referred to as TRPM2$_{DR\_ADPR/Ca2+}$). Our structural analyses identify unusual two-fold symmetric quaternary structural rearrangements in the channel assembly, which likely represent intermediate gating states. Moreover, a comparison with the published TRPM2$_{closed}$ and TRPM2$_{open}$ structures enabled us to speculate on the conformational pathway involving reduced symmetric rearrangements between the closed and open states of TRPM2.

## Results

### Structure determination and overall architecture of TRPM2$_{DR}$.
TRPM2$_{DR}$ shares ~50% sequence identity with human TRPM2 (Supplementary Fig. 1). When overexpressed in human embryonic kidney 293T (HEK293T) cells, wild-type TRPM2$_{DR}$ channels exhibit large currents in response to direct application of ADPR and $Ca^{2+}$ to the cytosolic side of inside-out patches (Supplementary Fig. 2). Notably, in the presence of saturating [$Ca^{2+}$] (125 μM), TRPM2$_{DR}$ shows an ADPR sensitivity similar to that of human TRPM2[24].

For structural characterization, we prepared the ligand-free TRPM2$_{DR}$ (TRPM2$_{DR\_Apo}$) sample in detergent. To determine structures in the ligand-bound conformations, TRPM2$_{DR}$ was purified in detergent in the presence of $Ca^{2+}$ (TRPM2$_{DR\_Ca2+}$) or subsequently reconstituted into amphipol in the presence of both $Ca^{2+}$ and ADPR (TRPM2$_{DR\_ADPR/Ca2+}$) (see "Methods"). Cryo-EM data were initially processed without imposing symmetry in order to avoid overlooking subpopulations of particles with reduced symmetry (C1 and C2). The appropriate symmetry was imposed to improve resolution of the reconstruction in the last stages of auto-refinement, after careful inspection of the asymmetric reconstructions confirmed the symmetry. The TRPM2$_{DR\_Apo}$, TRPM2$_{DR\_Ca2+}$, and TRPM2$_{DR\_ADPR/Ca2+}$ structures were finally determined to an overall resolution of ~4.3 Å, ~3.8 Å, and ~4.2 Å, respectively (Fig. 1 and Supplementary Figs. 3–5), enabling model building of ~70–84% of the TRPM2$_{DR}$ polypeptide (see "Methods", Supplementary Fig. 6 and Table 1). In our study, we observed density corresponding to two unique structural features of the TRPM2 channel: a second coiled-coil (CC2) and the NUDT9H domain at the C-terminus of the channel (Fig. 1). However, since these regions were the least well-defined in the EM maps, we facilitated model building and analysis by docking the homology model of the crystal structure of human ADPR pyrophosphatase NUDT9 (PDB 1Q33)[18] into the NUDT9H density (see "Methods").

Our TRPM2$_{DR}$ channel forms a homo-tetramer (Fig. 1a, b). Viewed orthogonally to the membrane plane, the channel can be divided into four layers. Like other published TRP channel structures, the TMD layer adopts a domain-swapped configuration between the VSLD and the pore[25,26]. Consistent with the previously reported TRPM2 structures[20,27,28], we also identified a $Ca^{2+}$ binding site located in the cavity formed by the VSLD in the TRPM2$_{DR\_Ca2+}$ and TRPM2$_{DR\_ADPR/Ca2+}$ structures (Supplementary Fig. 7a–c). While the TMD and two membrane-proximal CD layers resemble the architecture observed in the TRPM8 and TRPM4 structures[29–32], the additional CC2 and the NUDT9H domain together comprise a unique bottom CD layer in the TRPM2 structures (Fig. 1c). Similar to the previously reported TRPM2, TRPM4, and TRPM8 structures[20,27–32], each TRPM2$_{DR}$ protomer contains an N-terminal region composed of MHR1 to MHR4, a transmembrane channel region, and a C-terminal region. The CC2 serves as a link between the C-terminal NUDT9H domain and the rest of the channel (Fig. 1d).

### Symmetry analysis of the ligand-free TRPM2$_{DR}$ structure.
We prepared the TRPM2$_{DR\_Apo}$ according to the method published for the TRPM2$_{closed}$ structure. In brief, the protein was purified in $Ca^{2+}$-free buffer and incubated with 1 mM EDTA before sample vitrification (see "Methods"). The cryo-EM data were processed with no symmetry imposed during 3D reconstructions and classifications (Supplementary Fig. 3). Following Bayesian polishing, the shiny particles were subjected to a final round of 3D auto-refinement with C1 symmetry. This 3D reconstruction exhibited an overall apparent four-fold (C4) symmetry, but closer visual inspection revealed a slight deviation of the C4 symmetry

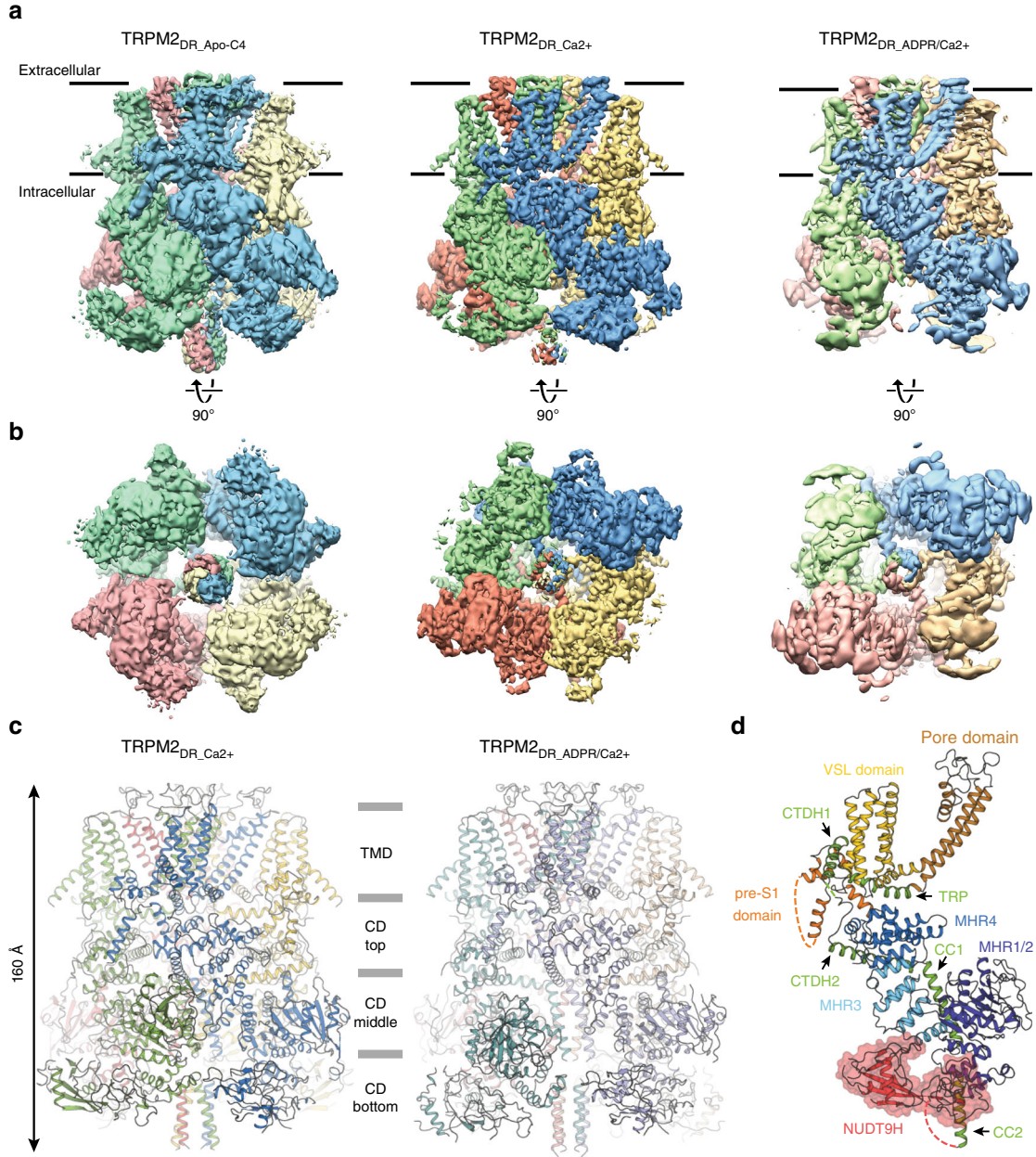

**Fig. 1** Overall architecture of TRPM2$_{DR}$ structures. Cryo-EM reconstructions of TRPM2$_{DR\_Apo-C4}$, TRPM2$_{DR\_Ca2+}$, and TRPM2$_{DR\_ADPR/Ca2+}$ structures viewed within the membrane bilayer (**a**) and from the cytosolic side (**b**). For TRPM2$_{DR\_ADPR/Ca2+}$, the NUDT9H domain is shown at 0.018 thresholding to emphasize conformational distinctions, while the rest of the molecule is shown at 0.033 thresholding. **c** Cartoon representations of the TRPM2$_{DR\_Ca2+}$ (left) and TRPM2$_{DR\_ADPR/Ca2+}$ (right) structures with the same orientation as in **a**. **d** Detailed view of the green-colored protomer of TRPM2$_{DR\_Ca2+}$ structure in **c**. The NUDT9H domain is highlighted with a red surface representation. Dashed lines indicate loops and helices that were not modeled in the structure

in the TMD. We therefore chose to proceed with the refinement in a conservative manner and refined the shiny particles with both C2 and C4 symmetry imposed in parallel and determined the reconstructions to ~4.5 Å and ~4.3 Å resolution, respectively (Supplementary Fig. 3). Our TRPM2$_{DR\_Apo}$ structure with imposed C4 symmetry (referred to as TRPM2$_{DR\_Apo-C4}$) resembles the published four-fold symmetric TRPM2$_{closed}$ structure across different layers of the channel (Supplementary Fig. 8a). At the single protomer level, the TRPM2$_{DR\_Apo-C4}$ structure superimposes well with the TRPM2$_{closed}$ structure (Fig. 2a, Cα RMSD = 1.5 Å).

To further assess the symmetry of our TRPM2$_{DR\_Apo}$ sample in detail, we built models separately into the C2 and C4 symmetric

maps. A comparison of the quaternary structural assemblies of these two models revealed similarities both in the TMD and across the CD layers (Supplementary Fig. 8b), suggesting that the C2-symmetric reconstruction contains four-fold symmetric elements (referred to as TRPM2$_{DR\_Apo-pseudo\ C4}$). However, upon closer inspection it became evident that, although the protomers A/C and B/D of the TRPM2$_{DR\_Apo-pseudo\ C4}$ structure retain similar overall conformations, slight deviations exist in both the pore domain and the CD (Fig. 2b; Cα RMSD = 1.99 Å). This suggests that these regions deviate from C4 symmetry, adopting a C2-symmetric organization within the tetramer. A direct comparison of individual protomers of the TRPM2$_{DR\_Apo-pseudo\ C4}$ structure and the TRPM2$_{DR\_Apo-C4}$ shows that protomers A/C

**Table 1 Data collection and refinement statistics**

| | TRPM2$_{DR\_Apo}$ | | TRPM2$_{DR\_Ca2+}$ | TRPM2$_{DR\_ADPR/Ca2+}$ |
|---|---|---|---|---|
| **Data collection** | | | | |
| Microscope | Titan Krios | | Talos Arctica | Titan Krios |
| Voltage (keV) | 300 | | 200 | 300 |
| Nominal magnification[a] | ×22,500 | | ×36,000 | ×59,000 |
| Exposure navigation | Image shift | | Image shift | Stage position |
| Total electron exposure (e$^-$ Å$^{-2}$) | 60 | | 63 | 40 |
| Exposure rate (e$^-$/pixel/s) | 15 | | 5.2 | 0.9 |
| Detector | Gatan K3 | | K2 Summit | Falcon 3EC |
| Pixel size (Å)[a] | 1.07 | | 1.15 | 1.39 |
| Defocus range (µm) | −0.75 to −2.5 | | −0.6 to −2.0 | −0.5 to −2.25 |
| Automation software | Latitude | | Leginon | EPU |
| Total extracted particles (no.) | 301,199 | | 1,791,114 | 1,949,622 |
| Refined particles (no.) | 184,457 | | 435,692 | 736,565 |
| **Reconstruction** | | | | |
| Final particles (no.) | 58,005 | | 93,573 | 135,120 |
| Symmetry imposed | C2 | C4 | C2 | C2 |
| Resolution (global) | 4.5 | 4.3 | 3.8 | 4.2 |
| FSC 0.5 (unmasked/masked) | 7.2/6.6 | 6.9/6.3 | 7.2/4.4 | 7.1/5.7 |
| FSC 0.143 (unmasked/masked) | 4.8/4.5 | 4.5/4.3 | 4.4/3.8 | 4.5/4.2 |
| Applied $B$-factor (Å$^2$) | −100 | −150 | −100 | −175 |
| **Refinement** | | | | |
| Protein residues | 4144 | 4348 | 4966 | 4848 |
| Ion | 0 | 0 | 4 | 2 |
| Ligand | 0 | 0 | 0 | 2 |
| Map Correlation Coefficient | 0.78 | 0.78 | 0.76 | 0.69 |
| $B$ factors (Å$^2$) | | | | |
| Protein residues | 221 | 165 | 101 | 89 |
| Ligands | N/A | N/A | 85 | 31 |
| **R.m.s. deviations** | | | | |
| Bond lengths (Å) | 0.007 | 0.005 | 0.007 | 0.006 |
| Bond angles (°) | 0.94 | 0.86 | 1.04 | 1.00 |
| **Ramachandran** | | | | |
| Outliers | 0.00% | 0.00% | 0.08% | 0.04% |
| Allowed | 5.82% | 4.68% | 6.42% | 7.55% |
| Favored | 94.18% | 95.32% | 93.50% | 92.41% |
| Poor rotamers (%) | 0.00% | 0.00% | 0.07% | 0.00% |
| MolProbity score | 1.58 | 1.42 | 1.79 | 1.72 |
| Clashscore (all atoms) | 4.0 | 3.08 | 6.64 | 4.79 |
| C-beta deviations | 0 | 0 | 0 | 0 |
| CaBLAM outliers (%) | 3.83% | 3.15% | 4.22% | 3.96% |

[a]Calibrated pixel size at the detector

(Cα RMSD = 0.96 Å) of the TRPM2$_{DR\_Apo-pseudo\ C4}$ align better to the TRPM2$_{DR\_Apo-C4}$ than the protomers B/D (Cα RMSD = 1.5 Å) (Fig. 2c). Based on our symmetry analysis, we speculate that the TRPM2 channel might possess an intrinsic propensity to undergo structural rearrangements that result in deviations from the canonical four-fold symmetry, even in the ligand-free condition. Due to the lower resolution of our TRPM2$_{DR\_Apo}$ structures, the published TRPM2$_{closed}$ structure was utilized for our detailed structural analyses of ligand-induced conformational changes.

**Structure in the intermediate state adopts two-fold symmetry.** In contrast to the canonical four-fold symmetry reported in the TRPM2$_{closed}$ and TRPM2$_{open}$ structures and the pseudo four-fold symmetry observed in the TRPM2$_{DR\_Apo}$ structure, our TRPM2$_{DR\_Ca2+}$ structure adopts pronounced two-fold symmetric arrangements when viewed along the central axis of the channel. This C2 symmetry was readily apparent even upon reference-free 2D classification of the particle images (Supplementary Fig. 4e). The salient features associated with the observed two-fold symmetry can be identified in the TRPM2$_{DR\_Ca2+}$ structure, which departs from the canonical four-fold symmetry to varying extents

across the layers of the channel. The symmetry reduction from C4 to C2 is most pronounced in the middle layer of the CD, which comprises the MHR1/2 and MHR3 domains (Fig. 3a and Supplementary Fig. 9), giving rise to conformationally distinct protomers within our TRPM2$_{DR\_Ca2+}$ structure. Notably, protomers A and C adopt a conformation that is consistent with the published TRPM2$_{closed}$ structure, while B and D resemble TRPM2$_{open}$ (Fig. 3b, c). This observation suggests that our TRPM2$_{DR\_Ca2+}$ structure may represent an intermediate state in which neighboring subunits adopt closed and open conformations in an alternating manner, resulting in the observed two-fold symmetry.

**Domain rearrangement at flexible junctions.** To identify the origin of the conformational divergence leading to our two-fold symmetric structure, we performed global and local alignments of protomers A and B in the TRPM2$_{DR\_Ca2+}$ structure. While the MHR3, MHR4, and VSLD are conformationally similar across protomers, substantial domain arrangements are observed at three junctions: NUDT9H-MHR1/2, MHR1/2-MHR3, and VSLD-pore (Fig. 4a, b). When the MHR3 and MHR4 domains of protomers A and B are aligned, the MHR1/2 and NUDT9H domains exhibit a drastic rotation along individual axes (Fig. 4c).

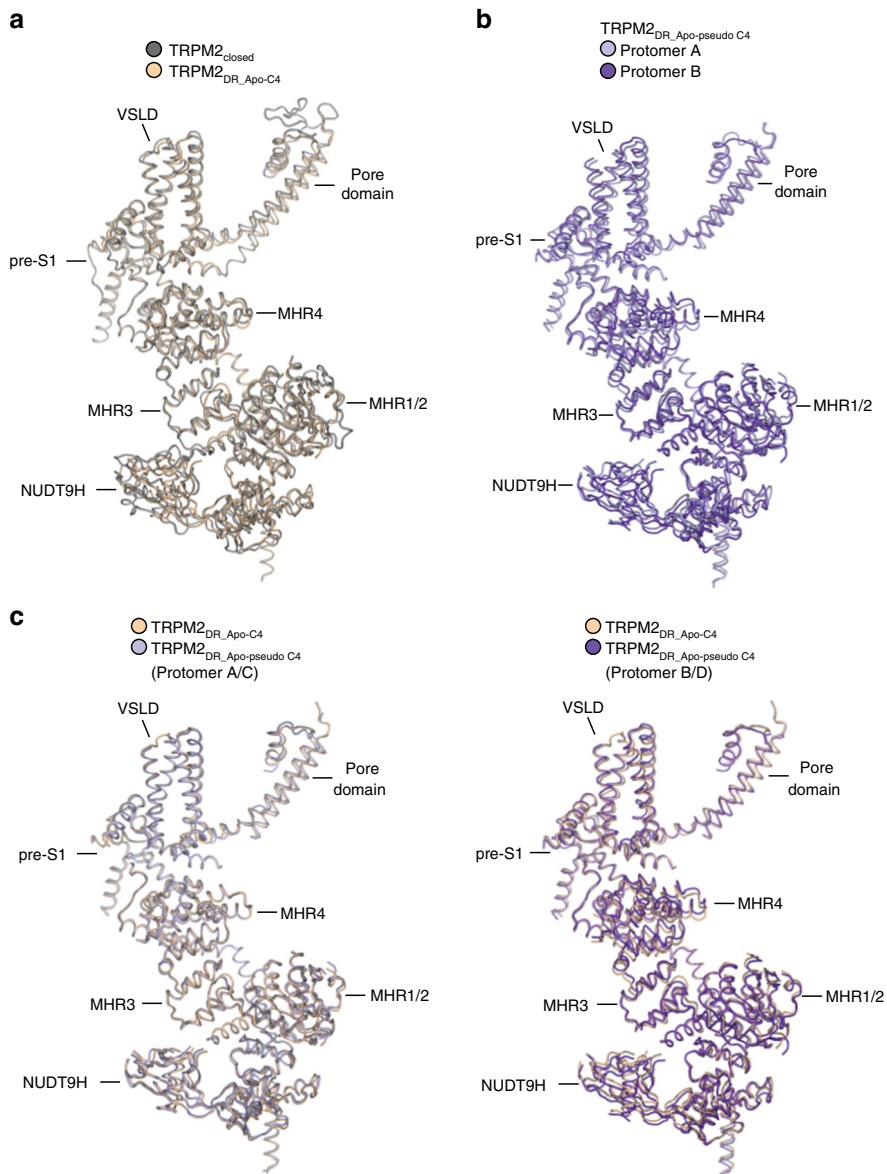

**Fig. 2** Comparison of TRPM2$_{DR}$ structures in the ligand-free condition. **a** Alignment (cartoon representation) of a protomer from the TRPM2$_{DR\_Apo-C4}$ structure (wheat) and from the TRPM2$_{closed}$ structure (silver, PDB 6DRK). **b** Alignment of protomers A (light purple) and B (purple) of the TRPM2$_{DR\_Apo-pseudo\ C4}$ structure. **c** Alignment of a protomer from the TRPM2$_{DR\_Apo-C4}$ structure (wheat) with protomers A/C (left, light purple) and B/D (right, purple) from the TRPM2$_{DR\_Apo-pseudo\ C4}$ structure, respectively

Moreover, this movement within the CD is propagated to the TM helices, resulting in two-fold symmetry within the TMD (Fig. 4d). The TMDs of the protomers A and B diverge at the S4b and the S4–S5 linker regions. In protomer A, the S4b adopts a 3$_{10}$-helical structure, while the equivalent region in protomer B contains an α-helical structure and an unstructured loop. In addition, the S4–S5 linker in protomer A contains a π-helix, while the corresponding region in protomer B is α-helical and forms a continuous straight helix with S5. Due to the absence of a bend-inducing π-helix at the junction between the S4–S5 linker and S5, the entire pore domain of protomer B is positioned differently from that of protomer A with respect to the central axis of the channel. Therefore, the flexible elements in the S4b and the S4–S5 linker lead to a two-fold symmetric configuration of the TMD. Notably, a similar two-fold symmetric TMD arrangement induced by a conformational change in the S4–S5 linker was recently observed in the TRPV2 channel[21,22]. Also, a ligand-dependent α-to-3$_{10}$ change in the secondary structure of S4b was

observed in TRPM8[33], which is in contrast to the 3$_{10}$-to-α transition in TRPM2. The substantial conformational change at the S4b and S4-S5 linker between adjacent protomers leads to unexpectedly distinct configurations of the S6 gate. Although the distances between diagonally opposed gate residues in TRPM2$_{DR\_Ca2+}$ are larger than those observed in the TRPM2$_{closed}$ structure, the S6 gate in TRPM2$_{DR\_Ca2+}$ is not as wide as that observed in the TRPM2$_{open}$ structure (Supplementary Fig. 10a). Taken together, the structural differences between protomers A and B in TRPM2$_{DR\_Ca2+}$ originate at the flexible junctions located between NUDT9H-MHR1/2, MHR1/2-MHR3, and VSLD-pore. This flexibility allows the channel to assume a two-fold symmetric intermediate state, where two subunits approximate the closed conformation and the other two adopt an arrangement that is similar to the open state.

Interestingly, the deviations from C4 symmetry that originate at the flexible junctions within the TRPM2$_{DR\_Ca2+}$ tetramer are also observed in the TRPM2$_{DR\_Apo-pseudo\ C4}$ structure. However,

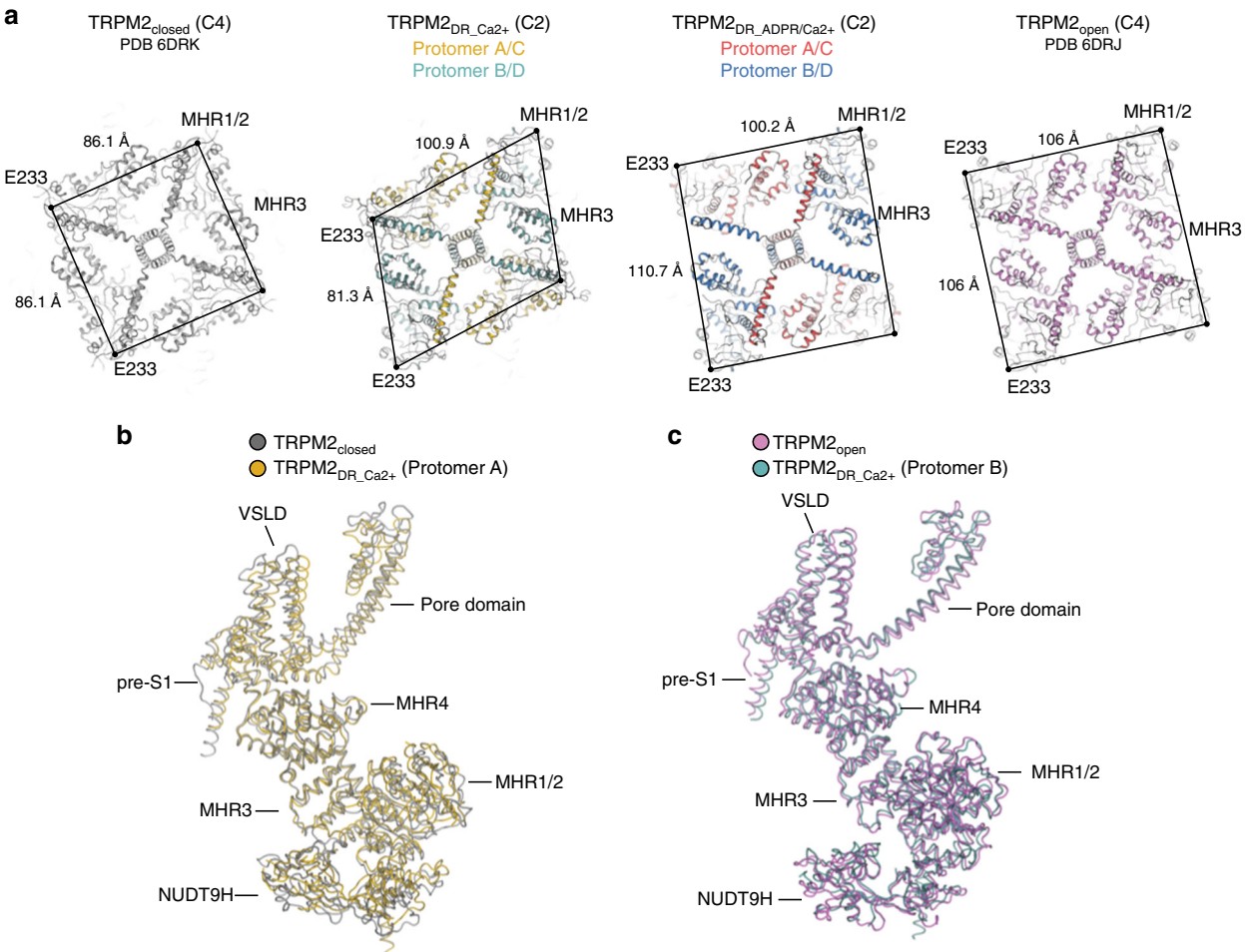

**Fig. 3** TRPM2$_{DR}$ structures in the intermediate states. **a** Extracellular views sliced through the middle layer of the CDs comprised of the MHR1/2 and MHR3 domains in the TRPM2$_{closed}$, TRPM2$_{DR\_Ca2+}$, TRPM2$_{DR\_ADPR/Ca2+}$, and TRPM2$_{open}$ structures. Structural alignment (cartoon representation) shows that protomer A in the TRPM2$_{DR\_Ca2+}$ structure (yellow) resembles the closed conformation in the TRPM2$_{closed}$ structure (silver, PDB 6DRK) (**b**), while the protomer B (teal) resembles the open conformation in the TRPM2$_{open}$ structure (violet, PDB 6DRJ) (**c**)

the extent of this deviation is much smaller and therefore results in a pseudo four-fold symmetry rather than the pronounced C2 symmetry observed in TRPM2$_{DR\_Ca2+}$ (Supplementary Fig. 11). When protomers A and B are aligned via their MHR3-4 domains, it becomes clear that the MHR1/2 and the NUDT9H domains rotate independently. Although both protomers A and B contain a helical S4–S5 linker at the junction between the VSLD and the pore domain, a slight divergence occurs at the S4b and the S4-S5 linker regions, which results in deviation from C4 symmetry in the TMD.

Importantly, the conformational differences between the closed and open states of the published TRPM2$_{DR}$ structures are similar to those between the two neighboring protomers in our TRPM2$_{DR\_Ca2+}$ structure (Supplementary Fig. 12). Given that protomers A and B in the TRPM2$_{DR\_Ca2+}$ structure resemble the closed and open conformations, respectively (Fig. 3b, c), we hypothesized that the addition of ADPR would induce further domain rearrangements at the junctions and enable transition of protomers A/C to the B/D configuration, converging the channel into a four-fold symmetric open conformation. To test this hypothesis, we prepared TRPM2$_{DR\_ADPR/Ca2+}$ by addition of ADPR to TRPM2$_{DR\_Ca2+}$ and determined its structure (Fig. 5). While protomer B in both structures retains an overall conformation that is similar to the open state (Fig. 5a), protomer A in TRPM2$_{DR\_ADPR/Ca2+}$ undergoes substantial

domain rearrangements in the CD at the interfaces between NUDT9H-MHR1/2 and MHR1/2-MHR3 (Fig. 5b). Notably, following the addition of ADPR, the CDs of protomers A and B in the TRPM2$_{DR\_ADPR/Ca2+}$ structure align well, which indicates that ligand binding changes the conformation of the CD in protomer A from the closed to the open state (Fig. 5c). Consequently, the CDs of the TRPM2$_{DR\_ADPR/Ca2+}$ structure converge to a nearly four-fold symmetric arrangement (Figs. 3a and 6).

**Alternating quaternary structure rearrangement in the CD.** Based on these structural comparisons, we suggest that the four subunits of TRPM2 channel adopt a two-fold symmetric intermediate quaternary structure assembly before ultimately assuming the canonical four-fold symmetric arrangement in the open conformation. To visualize the conformational changes associated with reduced symmetry, we compared our structures with the published TRPM2$_{closed}$ and TRPM2$_{open}$ structures at the bottom layer of the CD, which is comprised of the mobile MHR1/2 and NUDT9H domains, and identified substantial rearrangements mediated by critical interfacial interactions (Fig. 6).

It was shown that in the apo conformation, the NUDT9H domain in the TRPM2$_{closed}$ structure merely makes primary intra-subunit interactions with the MHR1/2 domain (Fig. 6a, d),

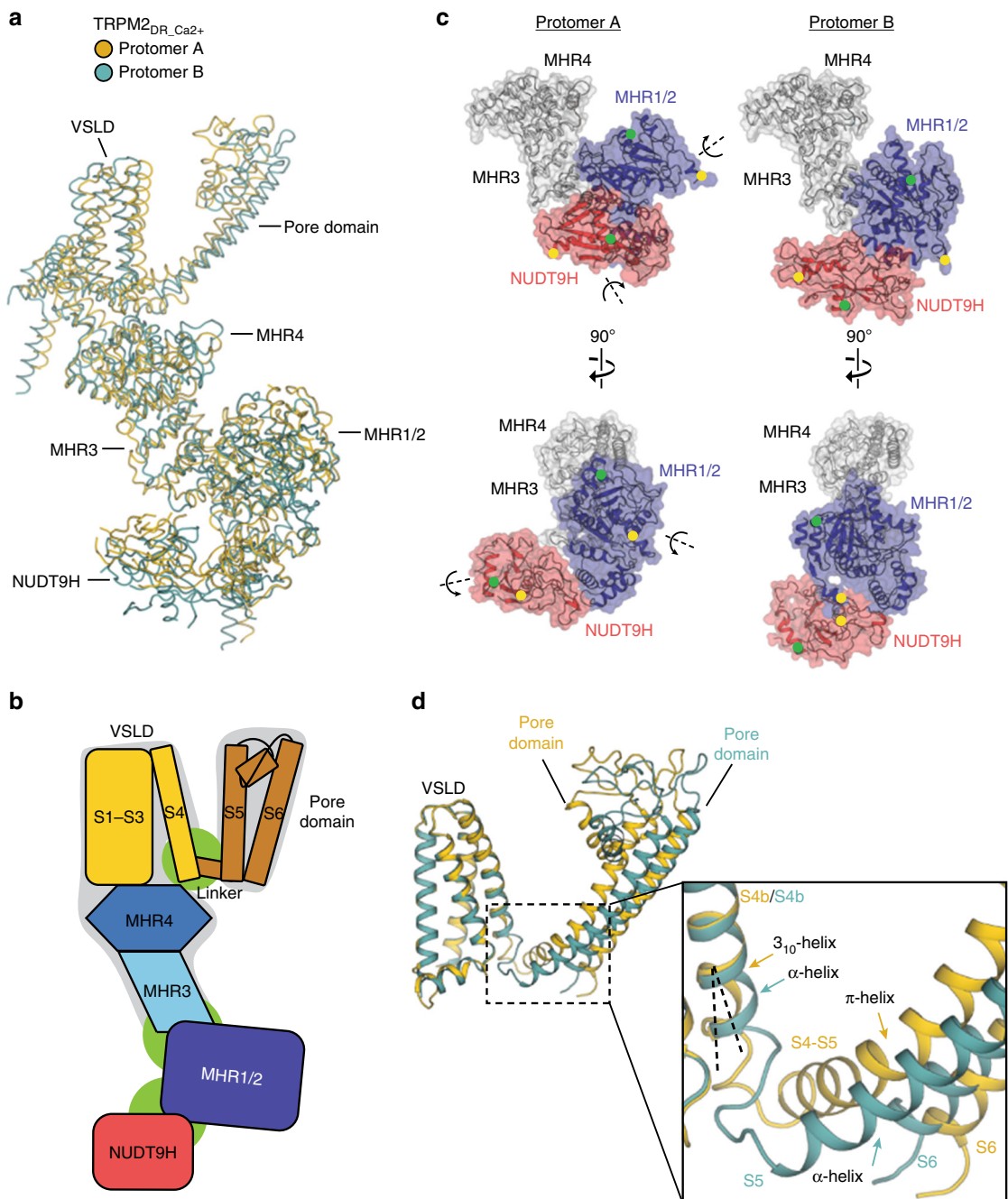

**Fig. 4** Flexible junctions enable conformational heterogeneity within the channel. **a** Cartoon representation showing alignment between protomer A (yellow) and protomer B (teal) from the TRPM2$_{DR\_Ca2+}$ structure. **b** Schematic diagram showing the three flexible junctions: NUDT9H-MHR1/2, MHR1/2-MHR3, and VSLD-pore (in green), and the static region: MHR3-MHR4-VSLD and the pore domain (in gray). The pre-S1 domain, TRP domain, and C-terminal helices are omitted for simplicity. **c** Protomers A (left) and B (right) of the TRPM2$_{DR\_Ca2+}$ structure aligned at MHR3 and MHR4 (gray). Side-by-side surface representations indicate that the rotations of MHR1/2 and NUDT9H in protomer A around individual axes (left) lead to the orientations observed in protomer B (right). D90 (yellow) and Q137 (green) from MHR1/2 and V1372 (green) and A1467 (yellow) from NUDT9H are denoted as dots in both protomers. **d** Protomers A (yellow) and B (teal) of the TRPM2$_{DR\_Ca2+}$ structure aligned at VSLD. Close-up view shows the structural divergence occurring at the S4b and the S4-S5 linker regions, giving rise to different configurations of the pore domain

while the binding of ADPR in the MHR1/2 domain relocates the NUDT9H domain to generate a secondary interface with the MHR1/2 domain from the neighboring protomer (Fig. 6c, g). Functional studies have shown that the interaction between the core subdomain of NUDT9H and the rest of the channel is important for stabilizing an open state of the TRPM2 channel, underscoring the importance of this secondary interface in channel opening[19]. Strikingly, our TRPM2$_{DR\_Ca2+}$ structure

adopts a combination of the two distinct interactions associated with the NUDT9H domain and the neighboring MHRs. Whereas the NUDT9H and MHR1/2 domains in protomer A exhibit only intra-subunit interactions, the core subdomain of the NUDT9H in protomer B makes additional contacts with the MHR1/2 domain in the neighboring protomer (Fig. 6b, e), resembling the interaction networks depicted in the TRPM2$_{closed}$ and TRPM2$_{open}$ structures, respectively. Furthermore, we observed changes in the

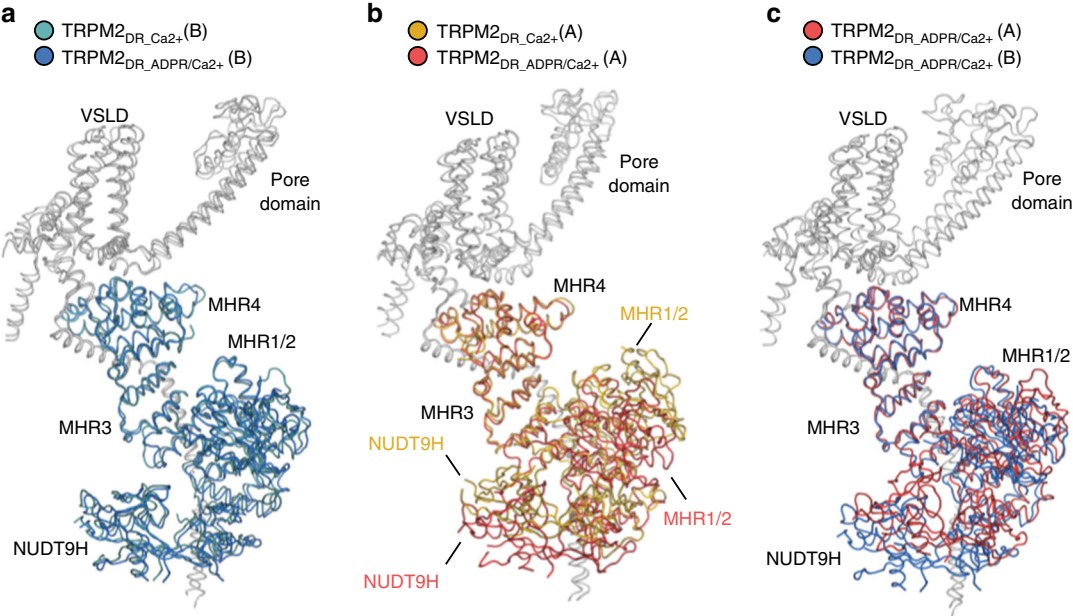

**Fig. 5** Addition of ADPR converts protomer configurations at the CD. Cartoon representations showing structural alignment of protomer B in the TRPM2$_{DR\_Ca2+}$ structure (teal) and protomer B in the TRPM2$_{DR\_ADPR/Ca2+}$ structure (blue) (**a**), protomer A in the TRPM2$_{DR\_Ca2+}$ structure (yellow) and protomer A in the TRPM2$_{DR\_ADPR/Ca2+}$ structure (red) (**b**), and protomer A (red) and protomer B (blue) in the TRPM2$_{DR\_ADPR/Ca2+}$ structure (**c**). Protomers are aligned at MHR4 domains. TMDs and C-terminal domains are colored in gray

interfacial interactions between the MHR1/2 domains and CC1. In the apo and closed state, the MHR1/2 domain is in loose contact with CC1 (Fig. 6d and Supplementary Fig. 13a). By contrast, in the open state, the binding of ADPR induces a displacement of the MHR1/2 domain, causing it to become disengaged from the CC1 helical bundle and positioned away from the central axis (Fig. 6g and Supplementary Fig. 13d). Our TRPM2$_{DR\_Ca2+}$ structure adopts both types of interactions between the MHR1/2 domain and CC1, further indicating a hybrid of the closed and the open conformations in the CD (Fig. 6e).

Therefore, based on the comparison with the TRPM2$_{closed}$ and TRPM2$_{open}$ structures, the alternating quaternary organization of the CDs in the TRPM2$_{DR\_Ca2+}$ structure suggests that ADPR-binding might induce further rearrangements that ultimately converge the reduced symmetry to adopt the four-fold symmetry apparent in the TRPM2$_{open}$ structure. In our TRPM2$_{DR\_ADPR/Ca2+}$ structure, we observe density for ADPR in the cleft of the MHR1/2 domain in protomers B and D, the shape and location of which resemble those depicted in the TRPM2$_{open}$ structure in complex with Ca$^{2+}$ and ADPR (Supplementary Fig. 7e). Although it is difficult to identify corresponding ADPR density in protomers A and C due to the low local resolution of the NUDT9H and MHR1/2 domains, we observed that the addition of ADPR converts protomers A/C from the closed to the open conformation, wherein the MHR1/2 domains detach from CC1 and the NUDT9H domains form interactions with neighboring MHR1/2 domains in all four protomers (Fig. 6f). As a result, the CDs in the TRPM2$_{DR\_ADPR/Ca2+}$ structure have been converted to a nearly four-fold symmetric arrangement that is similar to the CD conformation shown in the TRPM2$_{open}$ structure.

While its CDs adopt a nearly four-fold symmetric conformation, the TMDs of our TRPM2$_{DR\_ADPR/Ca2+}$ structure adopt a hybrid of the closed and the open conformations (Supplementary Fig. 10b–e), indicating that the TMDs have not converged on the open conformation. Similar to the TRPM2$_{DR\_Ca2+}$ structure, the diagonal distances at the S6 gate of the TRPM2$_{DR\_ADPR/Ca2+}$ structure are notably narrower than those in the published

TRPM2$_{open}$ structure in the open state. Furthermore, both our TRPM2$_{DR\_Ca2+}$ and TRPM2$_{DR\_ADPR/Ca2+}$ structures comprise two layers of aromatic residues lining the S6 gate, which may serve as a hydrophobic gate (Supplementary Fig. 10a). This suggests that the captured TRPM2$_{DR\_ADPR/Ca2+}$ structure likely represents a nonconductive intermediate state, wherein the structural rearrangements occurring at the CD have not been fully propagated to the TMD. We consider the possibility that the amphipol polymers used for the reconstitution of TRPM2$_{DR}$ in the presence of ADPR and Ca$^{2+}$ might tightly encircle the transmembrane regions and restrain transmission of structural rearrangements from the CD to the TMD, thereby preventing a two-fold- to four-fold-symmetric conformational rearrangement of the TMD. Recently, a similar hindrance to conformational changes in the TMD imposed by amphipol was discussed in the structural study of RTx-mediated activation of the TRPV2 channel[22].

**Subunit–subunit interfaces in the middle CD layer.** The interfaces between the MHR1/2 and MHR3 domains in adjacent subunits comprise the middle layer of the CD, which undergoes the largest conformational changes between the closed and open states upon ADPR binding. Notably, the reduced symmetry is most pronounced in the middle layer of the CD in the TRPM2$_{DR\_Ca2+}$ structure (Fig. 3a). Comparing the published TRPM2$_{closed}$ and TRPM2$_{open}$ structures showed that ADPR-induced conformational changes between MHR1/2 and MHR3 in each protomer leads to drastic changes in the subunit–subunit interface in the middle CD layer (Fig. 7a, c). While the closed state of TRPM2$_{DR}$ exhibits large interfacial interactions between neighboring MHR1/2 and MHR3 domains, the corresponding areas are substantially smaller in the open state (Fig. 7d, f). Notably, in the TRPM2$_{DR\_Ca2+}$ structure, the middle CD layer accommodates two distinct subunit–subunit interfaces (Fig. 7b, e), consistent with the most pronounced two-fold symmetry observed in this region (Fig. 3a). While the interface between MHR1/2 in protomer A and the neighboring MHR3 is similar to

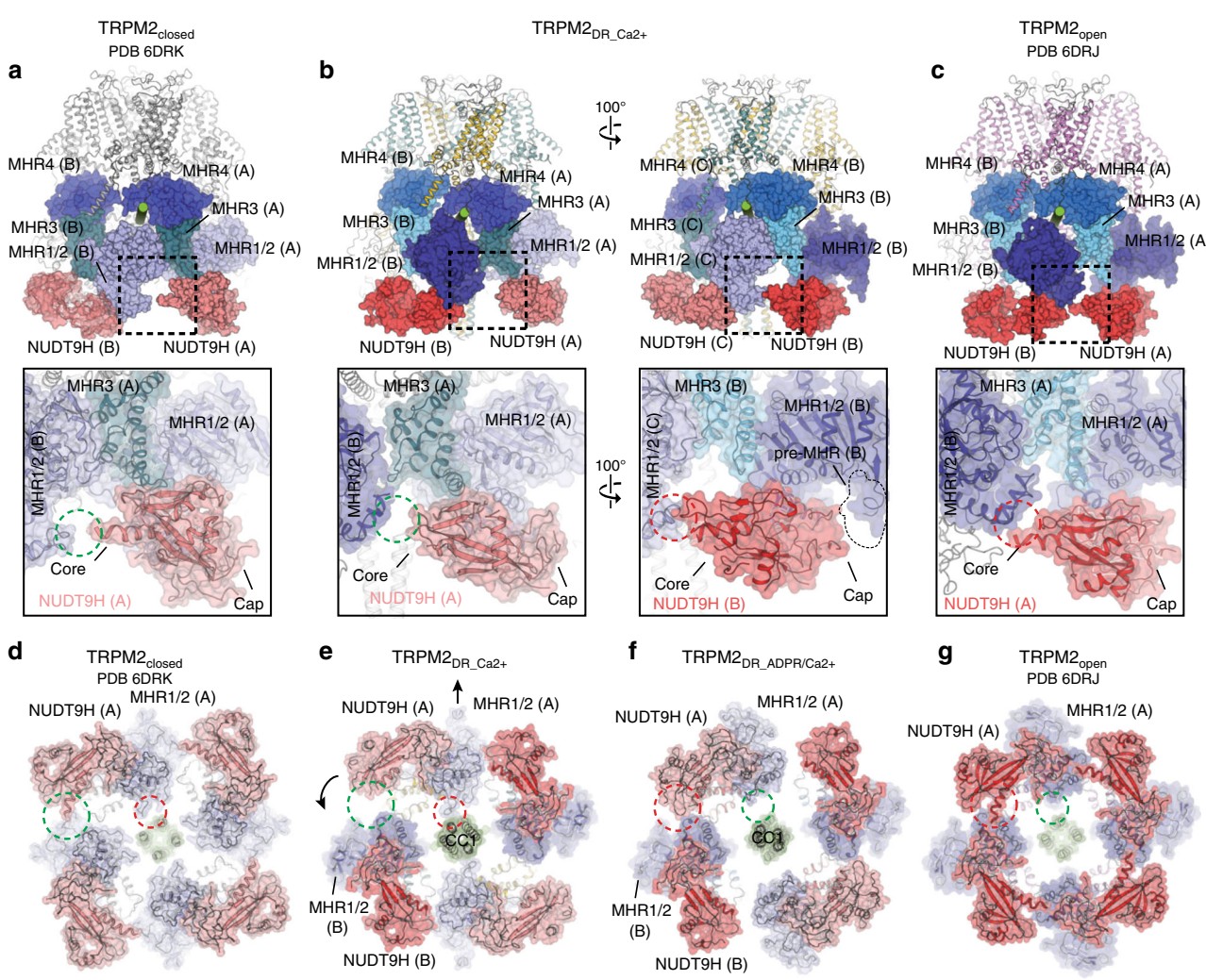

**Fig. 6** Alternating quaternary structure rearrangements in the cytoplasmic domain (CD). Comparison of the interaction networks between the NUDT9H domain and the neighboring MHR1/2 domain in the TRPM2$_{closed}$ (**a**, PDB 6DRK), TRPM2$_{DR\_Ca2+}$ (**b**), and TRPM2$_{open}$ (**c**, PDB 6DRJ) structures, respectively. Close-up views show that the NUDT9H domain in the TRPM2$_{closed}$ structure makes contact solely with MHR1/2 from the same subunit (**a**), while the core subdomain of the NUDT9H in the TRPM2$_{open}$ structure (**c**) makes additional interactions with the MHR1/2 from the adjacent protomer. More importantly, the TRPM2$_{DR\_Ca2+}$ structure adopts both interactions mediated by the NUDT9H and MHR1/2 domains (**b**). Viewed from the intracellular side, cartoon and transparent surface representations compare the conformational changes at the bottom layer of the CD in the TRPM2$_{closed}$ (**d**), TRPM2$_{DR\_Ca2+}$ (**e**) and TRPM2$_{DR\_ADPR/Ca2+}$ (**f**) from the current study, and TRPM2$_{open}$ (**g**) structures, respectively. Arrows indicate the domain movements observed in the TRPM2$_{DR\_Ca2+}$ (**e**) and TRPM2$_{DR\_ADPR/Ca2+}$ (**f**) structures relative to the channel in the closed conformation (**d**) and *en route* to the open conformation (**g**). Dashed circles highlight the detachment (green) and association (red) between the different domains as a result of structural rearrangements

the interfacial network in the closed state, the contact mediated by MHR1/2 in protomer B and the adjacent MHR3 domain resembles that of the open state. Our observation of these alternating interfaces in the TRPM2$_{DR\_Ca2+}$ structure prompted the question: "Why might a two-fold symmetric intermediate state exist?" We posit that concerted quaternary structural changes from the closed to the open state would invoke substantial rearrangements at the subunit–subunit interfaces, which could be energetically costly. However, by adopting an intermediate state with alternating subunit–subunit interfaces, the transition from the closed to the open state through a two-fold symmetric intermediate could reduce the energetic barrier for each structural rearrangement, thereby facilitating the channel gating (Fig. 7).

## Discussion

In this study, we examined the ligand-induced structural rearrangements in the TRPM2 channel. We determined the

TRPM2 structure in the apo conformation (TRPM2$_{DR\_Apo-C4}$), which resembles the published four-fold symmetric TRPM2$_{closed}$ structure, although upon further processing we observed a slight deviation from C4 symmetry in the TMD (TRPM2$_{DR\_Apo-pseudo\ C4}$). Notably, addition of Ca$^{2+}$ or both ADPR and Ca$^{2+}$ to TRPM2 (TRPM2$_{DR\_Ca2+}$ and TRPM2$_{DR\_ADPR/Ca2+}$, respectively) induces a conspicuous two-fold symmetric arrangement in the homo-tetrameric channel, a feature previously observed in the TRPV2 and TRPV3 channels[21–23]. Interestingly, a comparison of our structures with the recently published four-fold symmetric TRPM2$_{closed}$ and TRPM2$_{open}$ structures reveals that protomers A and C of TRPM2$_{DR\_Ca2+}$ resemble the closed conformation, while protomers B and D approximate the open conformation, indicating that the homo-tetramer can accommodate a hybrid of structural arrangements that are representative of the closed and the open states. This study effectively emphasizes the necessity of carefully examining the symmetry states of 3D reconstructions and applying symmetry in a conservative manner during cryo-EM data

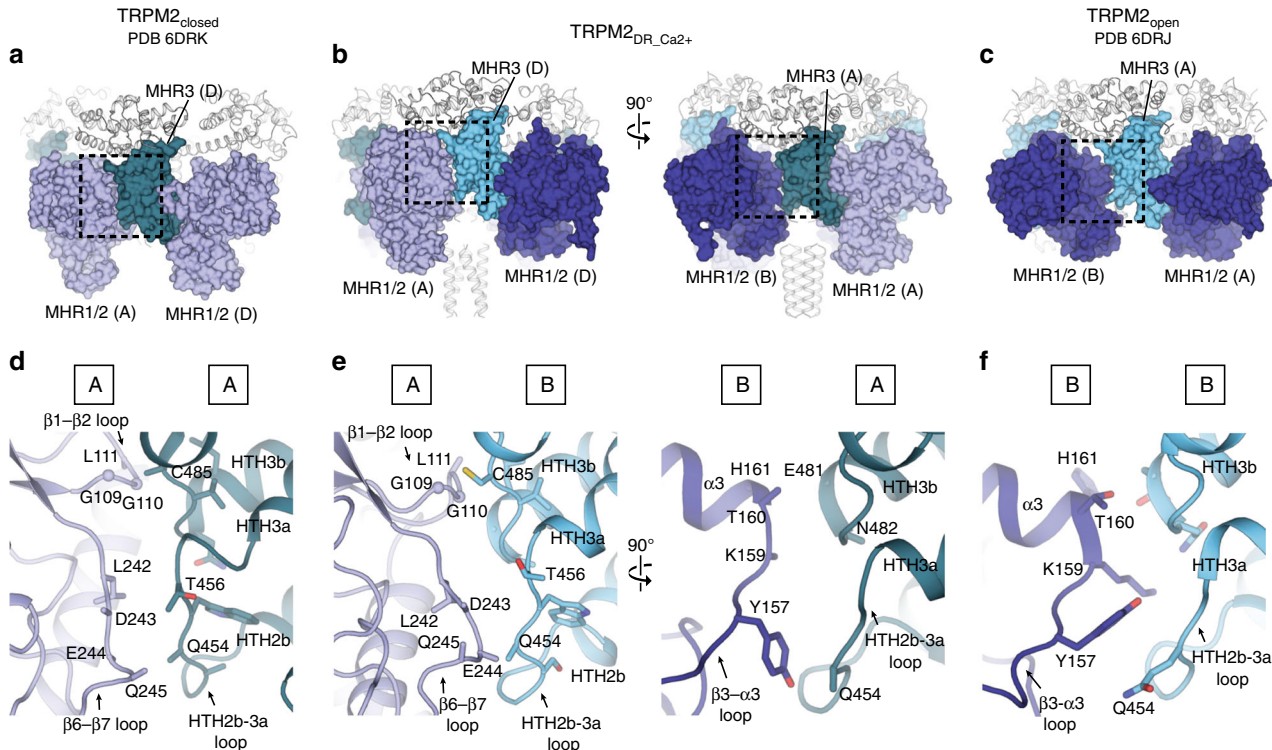

**Fig. 7** Subunit–subunit interfaces in the middle layer of CD. Comparison of the interaction networks between the neighboring subunits mediated by MHR1/2 and MHR3 domains in the TRPM2$_{closed}$ (**a**, PDB 6DRK), TRPM2$_{DR\_Ca2+}$ (**b**), and TRPM2$_{open}$ (**c**, PDB 6DRJ) structures. Close-up views of regions highlighted by dashed squares in **a–c** correspondingly. In the TRPM2$_{DR\_Ca2+}$ structure, the interface between MHR1/2 in protomer A and MHR3 in protomer D (equivalent to B) (**e**, left panel) resembles the interfacial network in the TRPM2$_{closed}$ structure (**d**); the interface between MHR1/2 in protomer B and MHR3 in protomer A (**e**, right panel) is similar to that in the TRPM2$_{open}$ structure (**f**)

processing, a strategy that has also been employed in the published studies of the magnesium channel CorA, TRPV2, and TRPV3 channels for capturing intermediate states[22,23,34] and of the mitochondrial calcium uniporter (MCU)[35–38]. It is noteworthy that among the published TRPM2 structures to date, either the C4 symmetry was imposed to the ab-initio map reconstructions from the beginning of data processing[28], or detailed descriptions of the stage for C4 symmetry imposition during data processing were lacking in the methods[20,27]. Therefore, in these published TRPM2 structures, all of which adopt the canonical four-fold symmetry, we cannot exclude the possibility that subclasses of particles with reduced symmetry (C1 or C2) might exist in their samples but could be overlooked upon application of the C4 symmetry at the early stage of data processing as a routine method.

Notably, our TRPM2$_{DR\_Ca2+}$ structure is the only structure of TRPM2 to date that was determined in the presence of Ca$^{2+}$ only, which adopt a unique two-fold symmetric arrangement. We consider two possibilities of this unusual quaternary structure: First, Ca$^{2+}$ binding in the VSLD confers flexibility to the junction between the VSLD and the pore (the S4–S5 linker), and thus primes the channel for opening. Consistent with this idea, the density for Ca$^{2+}$ in the TRPM2$_{DR\_Ca2+}$ reconstruction is stronger in protomer B than in protomer A (10.8 σ versus 9.5 σ, respectively) (Supplementary Fig. 7a, b). Second, endogenous ADPR happened to be captured in the Ca$^{2+}$-bound TRPM2 during sample preparation, as we also observe weak densities in the cleft of the MHR1/2 domain in protomers B and D of the TRPM2$_{DR\_Ca2+}$ reconstruction, which may correspond to ADPR (Supplementary Fig. 7d). The relatively poor quality of the density in these regions makes unambiguous assignment of these structural components difficult. Nevertheless, the potential presence of endogenous ADPR in this structure

suggests that ADPR might bind preferentially to two subunits first and result in a structure that is a hybrid of closed and open states. Although this second possibility does not support the role of Ca$^{2+}$ in inducing two-fold symmetric arrangement, it would still be in agreement with our proposed stepwise and induced-fit mechanism for ligand-dependent gating of TRPM2$_{DR}$.

Taken together, the alternating closed and open conformations of neighboring protomers observed in both our TRPM2 structures, as well as the conversion of the CD to a four-fold symmetric open conformation in the presence of additional ADPR, suggest that the TRPM2 channel adopts two-fold symmetric intermediate states *en route* to the open state (Fig. 8c). Reduced symmetry transitions could potentially be a mechanism employed to accommodate the substantial conformational changes that occur during channel activation. The subtle structural differences between protomers A and B in the TRPM2$_{DR\_Apo-pseudo\ C4}$ structure may also suggest an intrinsic flexibility at the domain interfaces and junctions, which disposes the channel toward symmetry transitions. Consistent with this idea, our interface analysis suggests that the two-fold symmetric intermediates confer an advantage by reducing the energetic barrier to the concerted quaternary structural changes going from the closed to the open state (Fig. 7). A similar departure from the canonical four-fold symmetry associated with TRP channel gating was recently observed during ligand-induced gating in the TRPV2 and TRPV3 channels[21–23], suggesting that adoption of a C2-symmetric conformation may be a more widely used mechanism in the TRP channel superfamily.

## Methods

**Protein expression and purification**. Zebrafish TRPM2 showed optimal biochemical stability based on a screen of eight orthologues including human and rat

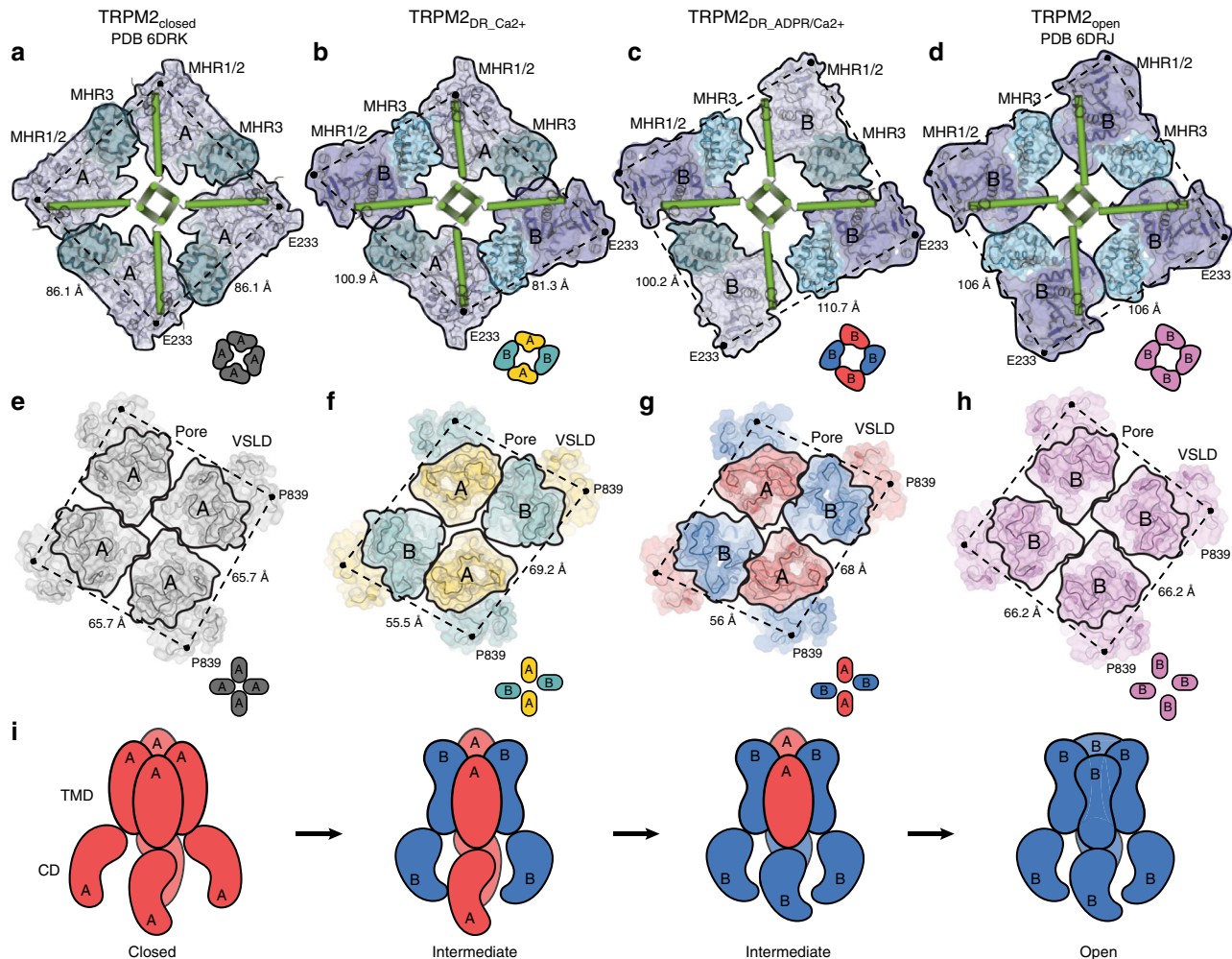

**Fig. 8** A trajectory of conformational changes during in the TRPM2$_{DR}$ channel gating. Viewed from the extracellular side, surface and cartoon representations of the middle layer of CD sliced through TRPM2$_{closed}$ (**a**), TRPM2$_{DR\_Ca2+}$ (**b**), TRPM2$_{DR\_ADPR/Ca2+}$ (**c**), and TRPM2$_{open}$ (**d**) structures. Individual protomers are highlighted in black frames. The protomer configurations are indicated by type "A" or "B" and also depicted as cartoon diagrams in insets. C$\alpha$ atoms of the E233 residues are shown as black dots, and distances between C$\alpha$ atoms are indicated (Å). Viewed from the extracellular side, surface and cartoon representations of the TMD of TRPM2$_{closed}$ (**e**), TRPM2$_{DR\_Ca2+}$ (**f**), TRPM2$_{DR\_ADPR/Ca2+}$ (**g**), and TRPM2$_{open}$ (**h**) structures. Individual pore domains are highlighted in black frames. The protomer configurations are indicated by type "A" or "B" and also depicted as cartoon diagrams in insets. C$\alpha$ atoms of the P839 residues are shown as black dots, and distances between C$\alpha$ atoms are indicated (Å). **i** Cartoon diagram of alternating quaternary structural rearrangements in TRPM2 channel gating

TRPM2. A codon-optimized and full-length gene for zebrafish (*Danio rerio*, XP_009303266.1) TRPM2 (TRPM2$_{DR}$) (Supplementary Table 1) was synthesized and custom cloned into a modified pEG BacMam vector[39] in frame with a C-terminal FLAG affinity tag (Bio Basic Inc.). The full-length wild-type protein was expressed by baculovirus-mediated transduction of HEK293S GnTI⁻ (*N*-acetylglucosaminyltransferase I-negative) suspension cells. In brief, the wild-type construct was transformed into DH10Bac *Escherichia coli* cells (ThermoFisher Scientific) and the recombinant bacmid was isolated and transfected into *Spodoptera frugiperda* (Sf9) cells (ATCC). The BacMam virus was generated, amplified, and added to HEK293S GnTI⁻ cells (ATCC) at a cell density of 3–3.5 M mL⁻¹. Cells were grown in Freestyle 293 media (Life Technologies) supplemented with 2% (v/v) FBS (Gibco) at 37 °C in the presence of 8% CO₂. After 18 h, 10 mM sodium butyrate was added to the cell culture and the temperature was reduced to 30 °C to boost protein expression. After 43–48 h of induction, cells were harvested and resuspended in buffer A (50 mM Tris pH 8, 150 mM NaCl, 1% digitonin (Sigma-Aldrich), 6.4 mM β-mercaptoethanol, 12 μg mL⁻¹ leupeptin, 12 μg mL⁻¹ pepstatin, 12 μg mL⁻¹ aprotinin, 1.2 mM phenylmethylsulfonyl fluoride and DNase I). Phytic acid (InsP₆, Sigma-Aldrich) was added to a final concentration of 1 mM. The protein was solubilized at 4 °C for 1 h. The insoluble material was removed by centrifugation at 8000 *g* for 30 min. The supernatant was incubated with anti-FLAG M2 resin (Sigma-Aldrich) for 30–40 min at 4 °C with gentle agitation. The resin was washed with ten column volumes of filtered buffer B (20 mM Tris pH 8, 150 mM NaCl, 0.06–0.1 % digitonin, 1 mM InsP₆, and 2 mM DTT) and eluted with five column volumes of the same buffer supplemented with 0.1 mg mL⁻¹ FLAG peptide. The elution was concentrated and further purified on a Superose 6

10/300 GL column (GE Healthcare) equilibrated with filtered buffer B. For cryo-EM analysis of the TRPM2$_{DR\_Apo}$ structure, the peak fractions were concentrated to 3.6 mg mL⁻¹ and incubated with 1 mM EDTA for 2 h before sample vitrification.

To prepare the sample for TRPM2$_{DR\_Ca2+}$ and TRPM2$_{DR\_ADPR/Ca2+}$ structures, the protein was solubilized and purified by anti-FLAG affinity chromatography and size-exclusion chromatography using the same procedures as described above, except that 2 mM CaCl₂ was included in buffer A and buffer B during purification. For the TRPM2$_{DR\_Ca2+}$ sample, peak fractions eluted from the size-exclusion column were concentrated to 0.5 mg mL⁻¹ for cryo-EM analysis.

To solve the TRPM2$_{DR\_ADPR/Ca2+}$ structure in complex with both ADPR and Ca²⁺, after purifying the protein by size-exclusion chromatography as described above, peak fractions were combined and mixed with Amphipol PMAL-C8 (Avanti polar lipid) at 1:10 (wt/wt) ratio and incubated overnight at 4 °C with gentle agitation. To remove detergent, 25 mg ml⁻¹ Bio-Beads SM-2 (Bio-Rad) was added and incubated at 4 °C for 1 h. After removing the Bio-Beads, the reconstituted protein was mixed with 50 μM ADPR (Sigma-Aldrich) and further purified on a Superose 6 10/300 GL column (GE Healthcare) equilibrated with buffer C (20 mM Tris pH8, 150 mM NaCl, 0.5 mM CaCl₂, 50 μM ADPR). The peak fractions were collected and concentrated to ~1.25 mg mL⁻¹ and incubated with ADPR to a final concentration of ~1 mM for 30–45 min before sample freezing for cryo-EM study.

**Cryo-EM sample preparation.** To prepare grids for the TRPM2$_{DR\_Apo}$ sample, 3 μL of protein in the presence of EDTA (3.6 mg mL⁻¹) was applied to freshly glow-discharged UltrAuFoil® R1.2/1.3 300-mesh grids (Quantifoil). The grids were

blotted using a Whatman No.1 filter paper for 3 s at 25 °C under 95% humidity before being plunged into liquid ethane using a Leica EM GP (Leica Microsystems).

For cryo-EM study of the TRPM2$_{DR\_Ca2+}$ structure, a thin amorphous carbon film was floated onto UltrAuFoil® R1.2/1.3 300-mesh grids (Quantifoil). Three µL of TRPM2 (0.5 mg mL$^{-1}$) was applied to grids that had freshly been plasma cleaned in a Solarus plasma cleaner (Gatan, Inc.) with a 75% argon/25% oxygen atmosphere at 15 W for 6 s. Grids were manually blotted[40] using a custom-built manual plunger in a cold room (≥95% relative humidity, 4 °C). Sample was blotted for ~4 s with a Whatman No.1 filter paper immediately prior to plunge freezing in liquid ethane cooled by liquid nitrogen.

The cryo-EM grids for the analysis of the TRPM2$_{DR\_ADPR/Ca2+}$ structure were prepared by applying 3 µL PMAL-C8 reconstituted TRPM2 (~1.25 mg mL$^{-1}$) in the presence of ADPR and Ca$^{2+}$ to freshly glow-discharged UltrAuFoil® R1.2/1.3 300-mesh grids (Quantifoil). The grids were blotted for 2 s with a Whatman Grade 1 filter paper at 25 °C under 95% humidity before being plunged into liquid ethane using a Leica EM GP (Leica Microsystems). The grids were stored in liquid nitrogen before data acquisition.

**Cryo-EM data acquisition and data processing.** For the TRPM2$_{DR\_Apo}$ structure, the cryo-EM data were collected on a Titan Krios (Thermo Fisher) TEM operating at 300 keV equipped with a K3 direct electron detector (Gatan, Inc) and acquired using the Latitude automated data-acquisition program. A total of 3776 movies (60 frames per movie) were acquired in counting mode at a nominal magnification of ×22,500 with a pixel size of 1.066 Å per pixel. Each movie was collected over a 4.6 s exposure with an exposure rate of about 15 e$^-$ pixel$^{-1}$ s$^{-1}$, resulting in a total exposure of about 60 e$^-$ Å$^{-2}$. A nominal defocus range was set from −0.75 to −2.5 µm. For data processing, motion correction and dose weighting were performed using MotionCor2[41], followed by CTF estimation of unweighted summed images using Gctf[42] in RELION-3.0. A total of 3635 micrographs were selected. A set of 1006 particles was manually picked and input to reference-free 2D classification ($k = 8$, $\tau = 2$), from which six classes were selected as templates for auto-picking of the entire dataset (301,199 particles). The particles were extracted, binned 4 × 4 (4.264 Å per pixel, 64-pixel box size) and input to reference-free 2D classification ($k = 30$, $\tau = 2$). 2D classes that showed clear TRPM2 channel features were combined (184,457 particles) and input to 3D auto-refinement with C1 symmetry using EMD-7127 low-pass filtered to 30 Å as an initial model. The auto-refined particles were recentered, reextracted binned 2 × 2 (2.132 Å per pixel, 128-pixel box size), and input to 3D-refinement with C1 symmetry. For the output 3D reconstruction, a soft mask (5 pixel extension, 5 pixel soft cosine edge) was created and used for 3D classification ($k = 2$, $\tau = 12$) of the refined particles without alignment. A set of 58,005 particles from one class which showed well-defined transmembrane and cytoplasmic domains was recentered, reextracted unbinned, and subject to 3D auto-refinement with C1 symmetry with a soft mask surrounding the full density. The refined particles were subject to Bayesian polishing followed by 3D auto-refinement without imposing symmetry (C1). Based on visual inspection, the 3D reconstruction exhibited an apparent four-fold symmetry with slight deviation in the pore domain. Therefore, the 3D auto-refinement job was repeated with two-fold symmetry (C2) and four-fold symmetry (C4) imposed separately in parallel, yielding a final construction of ~4.5 Å (C2) and ~4.3 Å (C4), respectively, determined by the gold-standard 0.143 Fourier shell correlation (FSC)[43]. Local resolution was calculated using RELION-3.0.

For the TRPM2$_{DR\_Ca2+}$ structure, the Cryo-EM data were acquired using the Leginon automated data-acquisition program[44]. All images preprocessing (frame alignment, CTF estimation, particle picking) was performed in real-time using the Appion image processing pipeline[45] during data collection. Images of frozen hydrated TRPM2 were collected on a Talos Arctica (Thermo Fisher) TEM operating at 200 keV. Movies were collected using a K2 Summit direct electron detector (Gatan, Inc) in counting mode at a nominal magnification of ×36,000 corresponding to a physical pixel size of 1.15 Å per pixel. A total of 3039 movies (64 frames per movie) of TRPM2 were collected by navigating to the center of four holes and image shifting ~2 µm to each exposure target. Movies were collected using a 16 s exposure with an exposure rate of 5.2 e$^-$ pixel$^{-1}$ s$^{-1}$, resulting in a total exposure of ~63 e$^-$ Å$^{-2}$ (1.17 e$^-$ Å$^{-2}$ per frame) and a nominal defocus range from −1.2 to −2 µm. The MotionCor2 frame alignment program[41] was used to perform motion correction and dose weighting as part of the Appion preprocessing workflow. Frame alignment was performed on 5 × 5 tiled frames with a B-factor of 100 applied. Unweighted summed images were used for CTF determination using CTFFIND4[46]. Difference of Gaussians (DoG) picker[47] was used to automatically pick particles from the first 636 dose-weighted micrographs yielding a stack of 176,961 particles that were binned 4 × 4 (4.6 Å per pixel, 80-pixel box size) and subjected to reference-free 2D classification using RELION 2.1[48]. The best nine classes were then used for template-based automated particle picking against the whole dataset using RELION. A total of 1,791,114 particles were extracted from these micrographs and binned 4 × 4 (4.6 Å per pixel, 80-pixel box size). Reference-free 2D classification in RELION was then used to sort out nonparticles and poor-quality picks in the data. A total of 435,692 particles corresponding to 2D class averages that displayed strong secondary-structural elements were input to 3D auto-refinement in RELION without symmetry imposed. EMD-7127 was low-pass filtered to 30 Å and used as an initial model. The refined particle coordinates were then used for recentering and reextraction of particles binned 2 × 2 (2.3 Å per pixel,

160-pixel box size). The resulting stack was subjected to 3D auto-refinement using the map obtained from the previous refinement as an initial model, and with a soft mask (5 pixel extension, 5 pixel soft cosine edge) generated from a volume contoured to display the full density. These particles were then subjected to 3D classification ($k = 6$, $\tau$ fudge = 12) without angular or translational searches using the same soft mask. Particles contributing to the classes that possessed the best-resolved densities around the transmembrane domain of the channel were 3D auto-refined and then recentered and reextracted without binning (135,215 particles, 1.15 Å per pixel, 320-pixel box size). These particles were then 3D auto-refined and subjected to 3D classification ($k = 3$, $\tau$ fudge = 12) without angular or translational searches. Two classes, comprising 94,028 particles, displayed the best-resolved density around the transmembrane region and C2 symmetry. 3D auto-refinement of these particles with C2 symmetry imposed yielded a ~3.9 Å reconstruction as determined by gold-standard 0.143 FSC[43], using phase-randomization to account for the convolution effects of a solvent mask on the FSC between the two independently refined half maps[49]. To improve map quality, all particles collected at greater than 2 µm defocus were removed. 3D auto-refinement of the final particle stack (93,573 particles) with C2 symmetry imposed yielded a ~3.8 Å reconstruction as determined by gold-standard 0.143 FSC.

For the TRPM2$_{DR\_ADPR/Ca2+}$ structure, the cryo-EM data were acquired using the EPU automated data-acquisition program. Images were collected on a Titan Krios operating at 300 keV equipped with a Falcon 3EC direct electron detector operating in counting mode. A total of 2496 movies were collected at a nominal magnification of ×59,000 with a physical pixel size of 1.39 Å per pixel using a nominal defocus range of −0.5 to −2.25 µm. Each movie (45 frames) was acquired using a dose rate of ~0.91 e$^-$ pixel$^{-1}$ s$^{-1}$ and a total exposure of ~40 e$^-$ Å$^{-2}$. Motion correction and dose-weighting were performed using the MotionCor2 frame alignment program on 5 × 5 tiled frames with a B-factor of 150 applied[41]. Gctf[42] was used for CTF estimation of unweighted summed images and 2426 good micrographs were selected. A set of 1719 particles were manually picked and subject to reference-free 2D classification ($k = 10$, $\tau = 2$) in RELION-3.0[50]. The best seven classes were used as templates for auto-picking of a total of 1,949,622 particles. The dataset was extracted unbinned and subjected to reference-free 2D classification, construction of ab-initio model, heterogeneous refinement, and homogeneous refinement in CryoSPARC[51], yielding a 3D reconstruction of ~4.3 Å resolution. In parallel, the 1,949,622 particles were extracted, Fourier binned 4 × 4 (5.56 Å per pixel, 64-pixel box size) and subjected to reference-free 2D classification. Good 2D classes showing secondary-structural features were selected. After further removing micrographs with a figure of merit below 0.1 and with astigmatism above 500 nm, a total of 736,565 particles from 2366 good micrographs were combined and input to 3D auto-refinement in RELION with C1 symmetry. The model of ~4.3 Å resolution generated by CryoSPARC was low-pass filtered to 30 Å and used as an initial model without a reference mask. The refined particles were recentered, reextracted, Fourier binned 2 × 2 (2.78 Å per pixel, 128-pixel box size), and subject to 3D auto-refinement with C1 symmetry, using the model from the previous auto-refinement as the reference mask and with a soft mask (5 pixel extension, 5 pixel soft cosine edge). The refined particles were input to 3D classification ($k = 4$, $\tau = 12$) without alignment using the same mask. A set of 135,120 particles comprising the best 3D class which shows that the most well-defined CD were recentered, reextracted, unbinned, and subject to 3D auto-refinement with C2 symmetry with a mask around the full density, yielding a final construction of ~4.2 Å resolution determined by gold-standard 0.143 FSC using RELION-3.0[50]. Per-particle CTF refinement and Bayesian polishing[52] were attempted, but the resolution was not improved.

**Model building and refinement.** An initial model of TRPM2 was generated by the RaptorX structure prediction server[53] with the sequence of zebrafish TRPM2 (XP_009303266.1). For the model building of the TRPM2$_{Ca2+}$ structure, individual domains (NUDT9H, VSLD, pre-S1, pore, CC1, CC2, MHR1/2, MHR3, and MHR4) were manually docked into the electron density map and the subsequent model building was performed manually in Coot[54]. A series of rigid-body fitting was performed for secondary structures within the structural domains, including the MHR domains, the pre-S1 domain, VSLD, the pore domain, and the NUDT9H domain. Residues with bulky side chains guided the correct register of helices and β-strands. Side chains were adjusted to optimal rotamer conformations and some loops that connect secondary structures were rebuilt to fit in the electron density. Unstructured loops and side chains whose electron density was not resolved in the map were deleted from the model. To facilitate the model building of the NUDT9H domain, the crystal structure of human ADPR pyrophosphatase NUDT9 (PDB ID: 1Q33) was docked into the electron density. The coordinates of the TRPM2$_{DR\_Ca2+}$ structure were docked into the cryo-EM map for the TRPM2$_{DR\_ADPR/Ca2+}$ structure, and subsequent model building was performed in a similar manner in Coot. For the TRPM2$_{DR\_Apo-C4}$ and TRPM2$_{DR\_Apo-pseudo\ C4}$ structures, the coordinates of protomer A of the TRPM2$_{DR\_Ca2+}$ structure were docked into both cryo-EM maps, followed by manual model building in Coot. Ideal geometry restraints were imposed on the secondary structure and the rotamer conformation as much as possible during the initial manual model building in Coot. In the TRPM2$_{DR\_Ca2+}$ structure, a calcium ion was placed into the EM density at the putative binding site in each protomer; while in the TRPM2$_{DR\_ADPR/Ca2+}$ structure, a calcium ion and an ADPR molecule were built into the putative EM density in protomers B and D.

The ligand restraints were generated by eLBOW[55]. No ligand molecules were modeled in the TRPM2$_{DR\_Apo-C4}$ or TRPM2$_{DR\_Apo-pseudo\ C4}$ structures. The four structure models were subsequently real-space refined in the PHENIX graphical interface against the cryo-EM map along with ligand restraints if any, using global minimization and rigid body refinement with secondary structure restraints[56]. Problematic regions in the real-space refined structure were identified using the Molprobity server (http://molprobity.biochem.duke.edu/)[57] and were manually fixed in Coot. FSCs of the half maps against the refined model agree with each other, indicating that the model is not over-refined (Supplementary Figs. 3–5). The final TRPM2$_{DR}$ structures cover about ~70–84% of the entire sequence. Poly-alanine models (UNK) were assigned for the CC2 (1188–1209) in all four structures.

**Electrophysiology**. HEK293T cells (62312975 – ATCC) were grown in DMEM supplemented with 10% FBS (Gibco), 1% penicillin/streptomycin (Gibco) and were sustained in 5% $CO_2$ atmosphere at 37 °C. Cells between passage 10–30 grown in 40-mm wells were transiently transfected at 30–50% confluency with plasmids encoding for TRPM2$_{DR}$ and green fluorescent protein using FuGene6 (Promega). Transfected cells were subcultured onto laminin (Sigma-Aldrich) coated 12-mm round glass coverslips (Fisher Scientific) 24 h post transfection and used 12–24 h after for electrophysiological measurements.

Voltage-clamp recordings were performed in the inside-out patch clamp configuration with glass electrodes pulled from borosilicate glass capillaries (Sutter Instruments) with a final resistance of 1.5–2.5 MΩ. The extracellular pipette solution contained (in mM) 140 NaCl, 5 KCl, 2 MgCl$_2$, 10 HEPES, 5 EGTA, and adjusted to pH 7.4 (NaOH). Glass coverslips were first placed in an open bath chamber (RC-26G, Warner Instruments) filled with the external pipette solution and after obtaining a GΩ seal, the patch was excised into the bath. An intracellular bath solution containing (in mM) 140 NaCl, 5 KCl, 2 MgCl$_2$, 10 mM HEPES, 1 EGTA, 1.12 CaCl$_2$ at pH 7.4 (NaOH), with estimated free $[Ca^{2+}]$ ≈125 μM calculated with MaxChelator software (http://maxchelator.stanford.edu)[58] with and without varying concentrations of ADPR (Sigma) (prepared daily from aqueous stock solutions (50 mM) stored at −80 °C) was applied to the inside of the patch with a pressurized perfusion system (BPS-8, ALA Scientific Instruments).

Inside-out current responses were elicited with a continuous repeating voltage ramp protocol (holding potential 0 mV for 50-ms before and after a 400-ms voltage ramp from 0 to +50 mV). Current responses were low-pass filtered at 1–2 kHz (Axopatch 200B), digitally sampled at 5–10 kHz (Digidata 1440 A), converted to digital files in Clampex10.7 (Molecular Devices), and stored on an external hard drive for offline analyses (Clampfit10.7, Molecular Devices; OriginPro 2016, OrginLab Corp.). The inward current at +50 mV ($V_m = -50$ mV) from each patch was used to calculate the mean current amplitude at each ADPR concentration.

**Reporting summary**. Further information on research design is available in the Nature Research Reporting Summary linked to this article.

## Data availability

Data supporting the findings of this manuscript are available from the corresponding author upon reasonable request. A reporting summary for this Article is available as a Supplementary Information file.

The source data underlying Supplementary Fig. 2 are provided as a Source Data file.

The sequence of TRPM2$_{DR}$ can be found in the National Center for Biotechnology Information under accession code XP_009303266.1. For the TRPM2$_{DR\_Apo-C4}$, TRPM2$_{DR\_Apo-pseudo\ C4}$, TRPM2$_{DR\_Ca2+}$, and TRPM2$_{DR\_ADPR/Ca2+}$ structures, the coordinates have been deposited in the Protein Data Bank with the PDB ID 6PKV, 6PKW, 6D73, and 6PKX and the cryo-EM density maps have been deposited in the Electron Microscopy Data Bank with the accession number EMD-20367, EMD-20368, EMD-7822, and EMD-20369.

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

## Acknowledgements

Cryo-EM data for TRPM2$_{DR\_Ca2+}$ were collected at The Scripps Research Institute (TSRI) electron microscopy facility. Cryo-EM data for TRPM2$_{DR\_Apo}$ and TRPM2$_{DR\_ADPR/Ca2+}$ were collected at the Duke Shared Materials Instrumentation Facility (SMIF). Preliminary cryo-EM work, including sample screening, was performed at the cryo-EM facility at NIEHS. We thank J.C. Ducom at the TSRI High Performance Computing facility for computational support and B. Anderson for microscope support. We thank L. Zubcevic, A. Bartesaghi, and L. Csanady for providing critical manuscript reading, developing a routine for preprocessing, and preliminary functional studies, respectively. This work was supported by the National Institutes of Health (R35NS097241 to S.-Y.L., DP2EB020402 and R21AR072910 to G.C.L.) and by the National Institutes of Health Intramural Research Program; US National Institute of Environmental Health Sciences (ZIC ES103326 to M.J.B). G.C.L is supported as a Searle Scholar, a Pew Scholar in the Biomedical Sciences, supported by the Pew Charitable Trusts, and by an Amgen Young Investigator award. M.W is supported by a National Science Foundation Graduate Student Research Fellowship. Computational analyses of EM data were performed using shared instrumentation funded by NIH S10OD021634.

## Author contributions

Y.Y. conducted all biochemical preparation, cryo-EM experiments and single-particle 3D reconstruction of TRPM2$_{DR\_Apo}$ and TRPM2$_{DR\_ADPR/Ca2+}$, and model building and refinement under the guidance of S.-Y.L. M.W. conducted cryo-EM experiments and single-particle 3D reconstruction of TRPM2$_{DR\_Ca2+}$ under the guidance of G.C.L. A.L.H. collected cryo-EM data of TRPM2$_{DR\_Apo}$ and TRPM2$_{DR\_ADPR/Ca2+}$ and helped with cryo-EM sample screening under the guidance of M.J.B. W.F.B. carried out electro-physiological recordings under the guidance of S.-Y.L. Y.Y., S.-Y.L., G.C.L. and M.W. wrote the paper.

## Additional information

**Competing interests:** The authors declare no competing interests.

