## [Peer Review File · Nature Communications]

Reviewers' Comments:

Reviewer #1:

Remarks to the Author:

Yin et. al. have performed a series of high quality experiments analyzing the structure of the zebrafish TRPM2. These results follow up on the recent reports of the open and closed TRPM2 structure for both zebrafish and human TRPM2 (Huang Y., 2018 and Wang L., 2018, resp.) What the authors are attempting to address here is the presentation of two intermediate states in the activation of drTRPM2. The addition of the activation mechanics described within would indeed be a very strong contribution to our understanding of TRP channels as similarly described for the TRPV family and worthy of publication, however the authors should focus on describing the novelty of these transitional states and describing their validation in more detail in light of the claims previously reported.

1) The authors claim (Fig 2) that the initial closed calcium bound state and ADPR/Ca bound state are quite distinct from previously reported structures with a unique two-fold symmetrical arrangement as opposed to the four-fold symmetry observed in the open and closed drTRPM2 and hsTRPM2 structures (PDB 6DRJ and 6DRK for drTRPM2). However, the authors utilize two different reconstitution strategies in determining their independent transition states. These methods include amphipol exchange for Ca and ADPR bound and detergent purification for Ca alone. Were these methods used to determine Apo and open states similar to the 4-fold symmetry previously reported or could it be the strategies employed in reconstitution and purification a cause of artifact? To address these concerns, the authors could state whether any of the 4-fold symmetrical states previously reported were observed in the large number of particles collected. Can the authors determine a comparable closed and open state using their existing data and if not, please explain or try and determine if a similar structure can be observed. Why was the exchange to PMAL-C8 necessary for the ADPR/Ca state? These methods are distinct from earlier reports and the current lack of the apo and four-fold symmetry open state using similar methods could help to alleviate some concerns that the transition states observed are purification and reconstitution dependent observations. A very large number of particles were excluded during the classification strategy. Over 1.7 million particles were used for each transition state. From here the number of particles excluded was over well over 1.6 million. What was the exclusion criterion used to determine the particular transition states and could this large scale exclusion be an explanation of how the purported calcium bound closed state and previously reported closed state differ so significantly?

In addition, the hsTRPM2 report claims ADPR binds to the NUDT9H domain where as the previously reported drTRPM2 and ADPR/Ca bound structure reported here points to the MHR1/2 domain as the location of ADPR binding. While ADPR could not be directly observed in the hsTRPM2 report, the authors should at the very least address these new findings in any new publication given that two reported binding sites have been observed and the drTRPM2- Δ NUDT9 lacked functional activity even though binding was not detected.

2) The major finding of this new report would be on the identification of two unique transitional states. Functional and biochemical validation should focus on this major point of the author's hypothesis. While the current functional analysis of TRPM2 expression in HEK cells was sufficient to show proper surface function/expression (although typically a negative control would be included in the data set for example R278A + R334A), the focus of this paper is on determination of the intermediate states. This represent the novelty of the authors claims and therefore requires functional validation. Could the authors utilize the information gained from their high-resolution transition structures and determine a method to test for the functional presence of these transition states in intact cells or with a more biochemical approach? Without evidence that the transitional state can be validated it is possible to interpret the new structures as an artifact of purification and reconstitution. This is especially true as the current methods used to isolate particles has not been previously reported for the ADPR/Ca open state and Ca-free closed state and should be introduced

in this paper. Further description of the physiological significance and validation of their possible role in channel activation should be addressed in more detail to warrant publication.

Reviewer #2:

Remarks to the Author:

The manuscript titled "Visualizing structural transitions of ligand-dependent gating of the TRPM2 channel" presents two TRPM2 channel structures, one in the presence of Ca²⁺ and another in the presence of both Ca²⁺ and ADP-ribose. These two structures were both solved by the cryoEM method with modest resolution, however, the authors went through a rigorous model building and refinement process. The final structures are compared to the two previously published structures, which represents the presumed closed state with no ligand (apo) and open state in the presence of both Ca²⁺ and ADP-ribose. Both structures in the current manuscript have a two fold symmetry, with the diagonally opposed pair of subunits assuming similar conformations. One pair of diagonally opposed subunits resembles the previous open conformation and another pair resembles the closed conformation. There are also regions where 3/10 and pi helices were observed along with several other differences to the previous structures. These structural analyses led to the conclusion that the current two structures represent two intermediate states in the closed to open process. The structure based conclusions are straightforward and I do not have much to add.

This field is extremely competitive, and at least two other groups have reported TRPM2 structures since the first two structures. The authors did their best given the structures and the limited time, and this work should be published.

Reviewer #3:

Remarks to the Author:

TRPM2 is a sensory cation channel that can be activated by simultaneous presence of intracellular Ca²⁺ and ADP-ribose. This manuscript by Yin et al reports zebrafish TRPM2 cryo-EM structures under two conditions: Ca²⁺ and ADPR plus Ca²⁺. The authors found C2 symmetry associated with the structures obtained under these conditions, in contrast to conventional C4 symmetry recently reported by three other groups for TRPM2 from *Nematostella vectensis*, zebrafish and human, respectively. This finding is interesting and potentially important for understanding conformational changes associated with channel activation and inactivation. The same research team recently found C2 symmetry in TRPV2 and -V3 as well; together they think there may exist some shared mechanism by TRPs in terms of the presence of non-canonical intermediate C2 states induced by ligand binding. The Discussion part can be enriched through responding to some of my comments as below.

Specific comments.

1. The authors stated that in their preparations Ca²⁺ was present, which they think resulted in C2 symmetry for Ca-bound condition. However, as there may be other differences, in addition to Ca²⁺, comparing with the conditions by other groups, it would be logically sound if they can exclude the potential contribution of those unknown differences to the formation of the C2 symmetry. Thus, I would hope that they can show that by removing Ca²⁺ they indeed will get a C4 symmetry (which also serves as a control); this will also be in agreement with the C4 symmetry found in the three existing papers (eLife, Nature and Science, 2018) and confirm that those unknown differences in experimental conditions are insignificant. This is important as it will indicate whether the intermediate, C2 symmetry state is mainly due to their special protein preparation conditions or it actually corresponds to a physiological state.

2. The structures in the presence of Ca plus ADPR obtained by other groups were called TRPM2_{open} while under the similar/same condition it is called TRPM2_{DR}-ADPR/Ca²⁺, which correspond to C₄ and C₂ symmetry, respectively. Although the C₂ under this condition is in fact quite close to a C₄ in terms of symmetry (Fig. 2a), the pore sizes are substantially different (Fig. S8a). I don't see which factor(s) within the experimental conditions that accounts for these differences. How would the author prepare open-state channel proteins that the other groups have obtained? Further, in Fig. S2, patch clamping data, would the TRPM2 proteins be in ADPR/Ca²⁺ C₂ symmetry state? If yes, then how to make the channels in the open, C₄ state, which should produce much larger currents due to a much bigger pore (Fig. S8a)? The authors may need to do some additional experiments address these questions while I ask them to discuss on all the differences and provide reasonable explanations. Further, by 'analogy', this triggers me to think that, perhaps under the same no Ca no ADPR closed state condition, the authors would obtain a structure/symmetry distinct from the published C₄. Even if this will be the case, I don't mean to say that it's incorrect; in contrast, to me this would make science more intriguing.

3. It would be interesting to examine the symmetry of ADPR-bound TRPM2 proteins (no Ca). A recent study using nvTRPM2 (eLife 2018) with ADPR alone (without intracellular Ca, but in the presence of extracellular Ca) can trigger partial channel activation. I would expect that the authors discuss about it and provide reasonable speculations to fit to their models (eg intermediate states and "stepwise mechanism" etc).

4. It seemed that they used 2 mM Ca²⁺ in their protein samples. I'm not sure whether this should correspond to intra- or extracellular Ca concentration when TRPM2 is on the cell surface membrane. If this is intracellular free Ca, 2 mM is extremely high and I would expect a much lower Ca concentration (eg in microM ranges) be used.

We thank the reviewers for appreciating the importance of our work. Their comments and criticisms have been helpful in improving our manuscript.

The reviewers have expressed concerns about the potential artifacts of the biochemical protein preparation methods that may have resulted in unexpected two-fold symmetric intermediate states of the TRPM2 channel presented in this study. We hereby include new TRPM2 apo structures showing four-fold symmetry comparable to the published results as an internal control along with relevant structural analyses in the revised manuscript. We provide the following point-by-point responses to the reviewers' comments.

Reviewer #1 (Remarks to the Author):

Yin et. al. have performed a series of high quality experiments analyzing the structure of the zebrafish TRPM2. These results follow up on the recent reports of the open and closed TRPM2 structure for both zebrafish and human TRPM2 (Huang Y., 2018 and Wang L., 2018, resp.) What the authors are attempting to address here is the presentation of two intermediate states in the activation of drTRPM2. The addition of the activation mechanics described within would indeed be a very strong contribution to our understanding of TRP channels as similarly described for the TRPV family and worthy of publication, however the authors should focus on describing the novelty of these transitional states and describing their validation in more detail in light of the claims previously reported.

1) The authors claim (Fig 2) that the initial closed calcium bound state and ADPR/Ca bound state are quite distinct from previously reported structures with a unique two-fold symmetrical arrangement as opposed to the four-fold symmetry observed in the open and closed drTRPM2 and hsTRPM2 structures (PDB 6DRJ and 6DRK for drTRPM2). However, the authors utilize two different reconstitution strategies in determining their independent transition states. These methods include amphipol exchange for Ca and ADPR bound and detergent purification for Ca alone. Were these methods used to determine Apo and open states similar to the 4-fold symmetry previously reported or could it be the strategies employed in reconstitution and purification a cause of artifact? To address these concerns, the authors could state whether any of the 4-fold symmetrical states previously reported were observed in the large number of particles collected. Can the authors determine a comparable closed and open state using their existing data and if not, please explain or try and determine if a similar structure can be observed

R) For the published zebrafish and human TRPM2 structures in the open and closed states, the protein samples were purified in detergent. By contrast, our TRPM2_{DR} Ca₂₊ was purified in detergent and our TRPM2_{DR} ADPR/Ca₂₊ was further reconstituted in amphipol. To test whether artifacts could have been introduced via purification or reconstitution, we have now determined the structure of TRPM2 structure in detergent in its ligand-free form (TRPM2_{DR} Apo) to serve as an internal control. This structure exhibits a pseudo four-fold symmetry comparable to the published TRPM2 structure in the closed state. In our revision, we have included this new data and have further considered and discussed the potential reasons for the existence of two-fold symmetry on pages 13 and 15.

Why was the exchange to PMAL-C8 necessary for the ADPR/Ca state? These methods are distinct from earlier reports and the current lack of the apo and four-fold symmetry open state using similar methods could help to alleviate some concerns that the transition states observed are purification and reconstitution dependent observations.

R) During optimization of the TRPM2_{DR ADPR/Ca2+} sample, we tried many different conditions for protein preparation and freezing. Here we found that, when prepared in detergent, the TRPM2_{DR ADPR/Ca2+} particles exhibited a tendency to form aggregates resulting in data which was unsuitable for high-resolution structure determination. Reconstituting the TRPM2_{DR ADPR/Ca2+} sample into PMAL-C8 amphipol alleviated the particle aggregation on grids. It is possible that the use of amphipol contributed to preferential capture of the intermediate state of TRPM2 where the conformational change in the CD is not propagated to TMD. A similar potential effect of amphipol was described in a recent study of TRPV2 (Zubcevic et al, eLife 2019). In our revision (on page 13), we have now included a discussion on the potential impact of amphipol on the conformational landscape of TRPM2_{DR}.

A very large number of particles were excluded during the classification strategy. Over 1.7 million particles were used for each transition state. From here the number of particles excluded was over well over 1.6 million. What was the exclusion criterion used to determine the particular transition states and could this large scale exclusion be an explanation of how the purported calcium bound closed state and previously reported closed state differ so significantly?

R) The input pool of particles (1.7M) included many damaged particles, false positives and contamination. Furthermore, both the TRPM2_{DR Ca2+} and the TRPM2_{DR ADPR/Ca2+} samples were heterogeneous. After extensive 2D and 3D classifications, we only selected the 3D classes that displayed the best resolved cryo-EM density in both the transmembrane domain (TMD) and the cytoplasmic domain (CD) including the NUDT9H domain. Only a small subset of the total pool of >1.7M particles made up these high-resolution 3D class(es). During 3D classification without symmetry imposed, we always visually examined the symmetry of the low-quality excluded classes, which adopted similar conformations as the selected 3D classes. In the case of TRPM2_{DR Ca2+}, strong two-fold symmetry was observed in the top/bottom views even during 2D classification without imposed symmetry. Because our new apo TRPM2 reconstruction, where we have used similar exclusion criteria, exhibits an apparent C4 symmetry, we do not think that the C2 symmetry in our Ca²⁺-bound conformation is likely to be caused by a classification artifact.

In addition, the hsTRPM2 report claims ADPR binds to the NUDT9H domain where as the previously reported drTRPM2 and ADPR/Ca bound structure reported here points to the MHR1/2 domain as the location of ADPR binding. While ADPR could not be directly observed in the hsTRPM2 report, the authors should at the very least address these new findings in any new publication given that two reported binding sites have been observed and the drTRPM2-ΔNUDT9 lacked functional activity even though binding was not detected.

R) We thank the reviewer for the suggestions. Because of the lack of clear cryo-EM density corresponding to ADPR in the human TRPM2 structure, we hesitated to address or make any comparison of the two distinct ADPR binding sites that have been reported up to date. In addition, our current study focuses on the structural transitions of the zebrafish TRPM2 channel during

ligand-dependent gating. Therefore, we chose to directly compare structures from the same species captured at different stages of the gating cycle and avoid any potential confusion (or over-interpretation) due to ortholog or species-dependent features of TRPM2 channels.

2) The major finding of this new report would be on the identification of two unique transitional states. Functional and biochemical validation should focus on this major point of the author's hypothesis. While the current functional analysis of TRPM2 expression in HEK cells was sufficient to show proper surface function/expression (although typically a negative control would be included in the data set for example R278A + R334A), the focus of this paper is on determination of the intermediate states. This represents the novelty of the authors' claims and therefore requires functional validation. Could the authors utilize the information gained from their high-resolution transition structures and determine a method to test for the functional presence of these transition states in intact cells or with a more biochemical approach? Without evidence that the transitional state can be validated it is possible to interpret the new structures as an artifact of purification and reconstitution. This is especially true as the current methods used to isolate particles has not been previously reported for the ADPR/Ca open state and Ca-free closed state and should be introduced in this paper. Further description of the physiological significance and validation of their possible role in channel activation should be addressed in more detail to warrant publication.

R) We understand the reviewer's requests for functional validation of the two-fold symmetric intermediate states described in our study. However, the TRPM2 channel exhibits a huge molecular weight of ~680kDa with 134 cysteine residues (34 cysteine residues per protomer), which makes biochemical studies to test the intermediate state challenging. More critically, because our C2 symmetric intermediate states contain interfaces that are present in both closed and open states, using functional studies to probe the transitional states is non-trivial (see Figs. 6 and 7). Taken together, it would be challenging to biochemically or functionally test the presence of intermediate states with reduced symmetry (e.g. via crosslinking experiments) during the revision period. Given the volume of work required to conduct such studies, we believe that they merit a separate, in-depth biochemical study.

Instead, we have now included additional structural data in this revised manuscript showing that the structure of the ligand-free TRPM2 (TRPM2_{DR_Apo}) adopts a four-fold symmetric conformation comparable to the published TRPM2 structures in the closed conformation (see responses in comment #1 for details). This serves as an internal control and demonstrates that no artifacts or biases towards two-fold symmetry were introduced during our biochemical preparation of the samples or our data processing methods. Importantly and interestingly, we would like to point out that although our apo TRPM2 reconstruction assumes a pseudo C4 symmetry and adopts a closed structure comparable to the published closed apo TRPM2 structure, our apo TRPM2 reconstruction exhibits slight departure from C4 symmetry. Upon closer inspection of the conformations of individual protomers within the apo TRPM2 channel refined with C2 symmetry imposed, we found that the slight departure from C4 symmetry is due to conformational changes at the flexible junctions which were also seen in the Ca²⁺ bound C2 symmetric TRPM2 structure. However, these conformational changes are much smaller in the apo than in the Ca²⁺-bound TRPM2. This observation has led us to speculate that TRPM2 channel might possess an intrinsic

propensity to undergo structural rearrangements that result in deviations from the canonical four-fold symmetry, even in the absence of ligands.

In addition, we would like to remind the reviewer that in the eLife and Nature reports of TRPM2 structures, the authors did not describe at which step(s) during the cryo-EM data processing they applied the C4 symmetry. However, in the Science paper, the authors imposed C4 symmetry to the ab initio map reconstructions from the beginning. Therefore, we cannot exclude the possibility that subclasses of particles with reduced symmetry (C1 or C2) exist in these published datasets but have been overlooked due to early application of C4 symmetry.

Reviewer #2 (Remarks to the Author):

The manuscript titled “Visualizing structural transitions of ligand-dependent gating of the TRPM2 channel” presents two TRPM2 channel structures, one in the presence of Ca²⁺ and another in the presence of both Ca²⁺ and ADP-ribose. These two structures were both solved by the cryoEM method with modest resolution, however, the authors went through a rigorous model building and refinement process. The final structures are compared to the two previously published structures, which represents the presumed closed state with no ligand (apo) and open state in the presence of both Ca²⁺ and ADP-ribose. Both structures in the current manuscript have a two fold symmetry, with the diagonally opposed pair of subunits assuming similar conformations. One pair of diagonally opposed subunits resembles the previous open conformation and another pair resembles the closed conformation. There are also regions where 3/10 and pi helices were observed along with several other differences to the previous structures. These structural analyses led to the conclusion that the current two structures represent two intermediate states in the closed to open process. The structure based conclusions are straightforward and I do not have much to add.

This field is extremely competitive, and at least two other groups have reported TRPM2 structures since the first two structures. The authors did their best given the structures and the limited time, and this work should be published.

R) We appreciate the reviewer’s comments.

Reviewer #3 (Remarks to the Author):

TRPM2 is a sensory cation channel that can be activated by simultaneous presence of intracellular Ca²⁺ and ADP-ribose. This manuscript by Yin et al reports zebrafish TRPM2 cryo-EM structures under two conditions: Ca²⁺ and ADPR plus Ca²⁺. The authors found C2 symmetry associated with the structures obtained under these conditions, in contrast to conventional C4 symmetry recently reported by three other groups for TRPM2 from *Nematostella vectensis*, zebrafish and human, respectively. This finding is interesting and potentially important for understanding conformational changes associated with channel activation and inactivation. The same research team recently found C2 symmetry in TRPV2 and -V3 as well; together they think there may exist some shared mechanism by TRPs in terms of the presence of non-canonical intermediate C2 states

induced by ligand binding. The Discussion part can be enriched through responding to some of my comments as below.

Specific comments.

1. The authors stated that in their preparations Ca^{2+} was present, which they think resulted in C2 symmetry for Ca-bound condition. However, as there may be other differences, in addition to Ca^{2+} , comparing with the conditions by other groups, it would be logically sound if they can exclude the potential contribution of those unknown differences to the formation of the C2 symmetry. Thus, I would hope that they can show that by removing Ca^{2+} they indeed will get a C4 symmetry (which also serves as a control); this will also be in agreement with the C4 symmetry found in the three existing papers (eLife, Nature and Science, 2018) and confirm that those unknown differences in experimental conditions are insignificant. This is important as it will indicate whether the intermediate, C2 symmetry state is mainly due to their special protein preparation conditions or it actually corresponds to a physiological state.

R) In the revised manuscript, we included TRPM2_{DR Apo} structures in the ligand-free condition and added a separate section on symmetry analysis in the Results section along with new figures (main Fig. 2 and supplementary Fig. 8).

As described on pages 6-7 and in the Methods section, the exclusion of ligands was performed in the same way as in the published TRPM2_{closed} structure, where protein was purified in Ca^{2+} -free buffer and incubated with 1mM EDTA before freezing. For cryo-EM data processing of the TRPM2_{DR Apo} structure in RELION, no symmetry (C1) was imposed until the last step of 3D auto-refinement. The 3D reconstruction adopts an apparent four-fold symmetry (C4). Closer visual inspection of the transmembrane domain revealed a slight two-fold symmetry (C2). Therefore, we applied both C2 and C4 symmetry to the final 3D reconstruction separately. In Fig. 2 and supplementary Fig. 8, our C4-symmetric structure (TRPM2_{DR Apo-C4}) resembles the published TRPM2_{closed} structure. The C2-symmetric apo structure is very similar to the TRPM2_{DR Apo-C4} structure, which indicates that it adopts a pseudo C4 symmetry. Taken together, by removing Ca^{2+} in the sample preparation, we are able to obtain TRPM2_{DR Apo} structures showing apparent C4 symmetry comparable to the published data.

Our new data serves as an internal control and negates the possibility that our biochemical preparation and structure determination have introduced artifacts that cause a systematic bias towards two-fold symmetry in our results. Importantly and interestingly, we would like to point out that although our apo TRPM2 reconstruction assumes a pseudo C4 symmetry and adopts a conformation comparable to the published closed apo TRPM2 structure, our apo TRPM2 reconstruction exhibits a slight departure from C4 symmetry. Upon closer inspection of the conformations of individual protomers within the apo TRPM2 channel refined with C2 symmetry imposed, we found that the slight departure from C4 symmetry is due to conformational changes at the flexible junctions which were also seen in the Ca^{2+} bound C2 symmetric TRPM2 structure. However, these conformational changes are much smaller in the apo than Ca^{2+} -bound TRPM2. This observation has led us to speculate that TRPM2 channel might possess an intrinsic propensity

to undergo structural rearrangements that result in deviations from the canonical four-fold symmetry, even in the absence of ligands.

In addition, we would like to remind the reviewer that in the eLife and Nature reports of TRPM2 structures, the authors did not describe at which step(s) during the cryo-EM data processing they applied the C4 symmetry. However, in the Science paper, the authors imposed C4 symmetry to the ab initio map reconstructions from the beginning. Therefore, we cannot exclude the possibility that subclasses of particles with reduced symmetry (C1 or C2) exist in these published datasets but have been overlooked due to early application of C4 symmetry.

2. The structures in the presence of Ca plus ADPR obtained by other groups were called TRPM2_{open} while under the similar/same condition it is called TRPM2_{DR}_ADPR/Ca²⁺, which correspond to C4 and C2 symmetry, respectively. Although the C2 under this condition is in fact quite close to a C4 in terms of symmetry (Fig. 2a), the pore sizes are substantially different (Fig. S8a). I don't see which factor(s) within the experimental conditions that accounts for these differences. How would the author prepare open-state channel proteins that the other groups have obtained? Further, in Fig. S2, patch clamping data, would the TRPM2 proteins be in ADPR/Ca²⁺ C2 symmetry state? If yes, then how to make the channels in the open, C4 state, which should produce much larger currents due to a much bigger pore (Fig. S8a)? The authors may need to do some additional experiments address these questions while I ask them to discuss on all the differences and provide reasonable explanations. Further, by 'analogy', this triggers me to think that, perhaps under the same no Ca no ADPR closed state condition, the authors would obtain a structure/symmetry distinct from the published C4. Even if this will be the case, I don't mean to say that it's incorrect; in contrast, to me this would make science more intriguing.

R) There are two major differences in the experimental conditions between our TRPM2_{DR}_ADPR/Ca²⁺ structure and the published TRPM2_{open} structure. Firstly, for the published TRPM2_{open} structure, both ADPR and Ca²⁺ were incubated together with the TRPM2 protein before freezing. In contrast, for our TRPM2_{DR}_ADPR/Ca²⁺ structure, Ca²⁺ was included in the buffer from the beginning of the purification, while ADPR was introduced at a later stage during amphipol reconstitution. The binding of Ca²⁺ might prime the channel to open in a two-fold symmetric conformation before ADPR was included. Secondly, although in both cases TRPM2 channel was solubilized and purified in detergent, our TRPM2_{DR}_ADPR/Ca²⁺ sample was further reconstituted into amphipol PMAL-C8. As discussed on page 13, we consider the possibility that the amphipol polymer might tightly constrict the transmembrane domains (TMD) of the channel, thereby prohibiting the propagation of structural rearrangements from the cytoplasmic domain (CD), which already nearly converges into four-fold symmetry, to the TMD. As a result, the TMD of our TRPM2_{DR}_ADPR/Ca²⁺ structure is trapped in a two-fold symmetric configuration. Our speculation is in line with a recent study of RTx-bound TRPV2 which compared the conformational space that can be accessed by the channel reconstituted in either amphipol or nanodiscs (Zubcevic et al, eLife 2019). To achieve the open-state structure, we would keep the protein in detergent and follow the protocol for the published open TRPM2_{DR} structure. Because we reproduced the published closed TRPM2_{DR} structure, we believe that we would also be able to reproduce the open structure of TRPM2_{DR}. However, because we have focused on determining a

new structure of the ligand-free TRPM2_{DR} during our revision in response to the reviewer's request, obtaining and reproducing the open state of the TRPM2_{DR} structure is out of the scope of the current study.

Regarding the relationship between the magnitude of channel current and the different functional states of the pore, we are afraid that determining the symmetry of the TRPM2 pore based on the relative current size would be very difficult and fraught with uncertainties.

Although the S6 gate is relatively wide in our intermediate states, two layers of pore-lining aromatic residues will serve as a hydrophobic gate. This leads us to think that these intermediate states represent non-conductive conformations of TRPM2 (supplementary Fig. 10). Therefore, we believe that only the published C4 symmetric ADPR/Ca²⁺-bound TRPM2_{DR} represents an open channel. In order to address the functional states of our structures, we included additional discussion on page 13 of the revised manuscript ("Similar to the TRPM2_{DR_Ca2+} structure.... fully propagated to the TMD).

Regarding the reviewer's thoughts on the "no Ca no ADPR closed state condition", please refer to our responses in comment #1, where we discussed the addition of the new TRPM2_{DR_Apo} structure in the ligand-free condition to the revised manuscript. As discussed above, we observed slight departure from C4 symmetry in the ligand-free TRPM2_{DR} (no Ca²⁺ and no ADPR) structure, which is due to conformational changes at the flexible junctions that were also seen in the Ca²⁺-bound two-fold symmetric TRPM2 structure, albeit to a much smaller extent. This observation has led us to speculate that the TRPM2_{DR} channel might possess an intrinsic propensity to undergo structural rearrangements that result in deviations from the canonical four-fold symmetry, even in the absence of ligands.

3. It would be interesting to examine the symmetry of ADPR-bound TRPM2 proteins (no Ca). A recent study using nvTRPM2 (eLife 2018) with ADPR alone (without intracellular Ca, but in the presence of extracellular Ca) can trigger partial channel activation. I would expect that the authors discuss about it and provide reasonable speculations to fit to their models (eg intermediate states and "stepwise mechanism" etc).

R) We understand the reviewer's suggestion that including the ADPR only-bound TRPM2 structure as another intermediate state would complement the path of ligand-gated activation of TRPM2 channel. However, we have focused on determining a new structure of the ligand-free TRPM2_{DR} during our revision in response to the reviewer's request. It would cost a significant amount of resource and time to determine another structure. The new TRPM2_{DR_Apo} structures introduced in the revised manuscript serve more essential and critical roles in validating our biochemical preparation and structural determination. The ADPR only-bound structure, while certainly a valuable addition, would be less critical within the scope of the current study.

4. It seemed that they used 2 mM Ca²⁺ in their protein samples. I'm not sure whether this should correspond to intra- or extracellular Ca concentration when TRPM2 is on the cell surface membrane. If this is intracellular free Ca, 2 mM is extremely high and I would expect a much lower Ca concentration (eg in microM ranges) be used.

R) TRPM2 channel is regulated by intracellular calcium. For the TRPM2_{DR_Ca2+} and TRPM2_{DR_ADPR/Ca2+} structure determination, we included a high concentration of Ca²⁺ in the buffer during purification, in order to saturate the Ca²⁺ binding site in the channel. Using a mM

concentration of Ca^{2+} to saturate the Ca^{2+} binding site has been routinely employed in many structural studies of Ca^{2+} binding proteins. For example, in both published TRPM2 structures 1 mM Ca^{2+} was used (Huang et al, 2018; Wang and Fu et al, 2018); in our recent TRPM8 studies, we used 2 mM Ca^{2+} (Yin et al, 2019). Also, the studies of calmodulin binding to the C-terminal domain of the Na_v channels typically use 1-2 mM Ca^{2+} , although the affinity of calmodulin for Ca^{2+} is much higher (Wang et al, Nat Commun, 2014). Therefore, we do not think that the high concentration of Ca^{2+} used is likely to be the reason for the observed two-fold symmetry.

Reviewers' Comments:

Reviewer #1:

Remarks to the Author:

This referee would like to commend the efforts and communication from the authors and truly appreciate the time they spent responding and improving the paper. A significant improvement is noted and many of the concerns are well addressed. Potential caveats leading to potential misinterpretation, which can occur in any paper, have been clearly discussed in the body and discussion of the revised manuscript. Therefore, this manuscript should be published in Nature Communications.

Reviewer #3:

Remarks to the Author:

The authors have performed new experiments based on my major concerns or comments and satisfactorily addressed or discussed those issues. I have no more comment on the revised manuscript which has been much improved.